# Iron imaging in myocardial infarction reperfusion injury

Brianna F. Moon [1], Srikant Kamesh Iyer [2], Eileen Hwuang [1], Michael P. Solomon[1], Anya T. Hall[1], Rishabh Kumar[3], Nicholas J. Josselyn[2], Elizabeth M. Higbee-Dempsey[4], Andrew Tsourkas [1], Akito Imai[5], Keitaro Okamoto[5], Yoshiaki Saito[5], James J. Pilla[2], Joseph H. Gorman III[5], Robert C. Gorman[5], Cory Tschabrunn[6], Samuel J. Keeney[5,7], Estibaliz Castillero [8], Giovanni Ferrari[8], Steffen Jockusch[9], Felix W. Wehrli[2], Haochang Shou[10], Victor A. Ferrari[6], Yuchi Han[6], Avanti Gulhane[2], Harold Litt[2], William Matthai[11] & Walter R. Witschey [1,2,4 ✉]

Restoration of coronary blood flow after a heart attack can cause reperfusion injury potentially leading to impaired cardiac function, adverse tissue remodeling and heart failure. Iron is an essential biometal that may have a pathologic role in this process. There is a clinical need for a precise noninvasive method to detect iron for risk stratification of patients and therapy evaluation. Here, we report that magnetic susceptibility imaging in a large animal model shows an infarct paramagnetic shift associated with duration of coronary artery occlusion and the presence of iron. Iron validation techniques used include histology, immunohistochemistry, spectrometry and spectroscopy. Further mRNA analysis shows upregulation of ferritin and heme oxygenase. While conventional imaging corroborates the findings of iron deposition, magnetic susceptibility imaging has improved sensitivity to iron and mitigates confounding factors such as edema and fibrosis. Myocardial infarction patients receiving reperfusion therapy show magnetic susceptibility changes associated with hypokinetic myocardial wall motion and microvascular obstruction, demonstrating potential for clinical translation.

[1] Department of Bioengineering, University of Pennsylvania, Philadelphia, PA, USA. [2] Department of Radiology, Perelman School of Medicine, University of Pennsylvania, Philadelphia, PA, USA. [3] Department of Biophysics, University of Pennsylvania, Philadelphia, PA, USA. [4] Biochemistry and Molecular Biophysics Graduate Group, Perelman School of Medicine, University of Pennsylvania, Philadelphia, PA, USA. [5] Department of Surgery, Perelman School of Medicine, University of Pennsylvania, Philadelphia, PA, USA. [6] Department of Medicine, Perelman School of Medicine, University of Pennsylvania, Philadelphia, PA, USA. [7] Department of Pediatrics, Children's Hospital of Philadelphia, Philadelphia, PA, USA. [8] Department of Surgery, Columbia University Irving Medical Center, New York, NY, USA. [9] Department of Chemistry, Columbia University, New York, NY, USA. [10] Department of Biostatistics, Epidemiology and Informatics, Perelman School of Medicine, University of Pennsylvania, Philadelphia, PA, USA. [11] Department of Medicine, Penn Presbyterian Medical Center, University of Pennsylvania, Philadelphia, PA, USA. ✉email: witschey@pennmedicine.upenn.edu

Reperfusion injury is a complication of vascular reperfusion therapy for acute myocardial infarction and is estimated to occur in up to 60% of patients[1,2]. While reperfusion remains the most effective strategy for reducing infarct size and substantially improves myocardial salvage, reperfusion injury may be independently associated with adverse left ventricular (LV) remodeling, increased risk of fatal arrhythmias and hospitalization for heart failure[1,3–11].

These findings highlight the need for noninvasive imaging techniques, specific to the molecular components of reperfusion injury, to identify patients at high risk for adverse LV remodeling and heart failure. High-risk patients could be treated more aggressively, while allowing low-risk patients to be seen less frequently for follow up, improving cost effectiveness. Patients could be enrolled in studies of emerging preventative approaches such as stem cell and biomaterial therapy[12–14]. Imaging could be used to evaluate the effectiveness or the dose optimization of adjunctive therapies for reperfusion such as periprocedural glycoprotein IIb/IIIa inhibitor[15], iron chelation therapy[16], and β1-adrenergic receptor antagonists[17].

While iron is an essential biometal in many normal metabolic reactions involving cardiac function, it also has a pathologic role in reperfusion injury[8,18], specifically the production of reactive oxygen species, and subsequent DNA damage. Magnetic resonance imaging (MRI) provides a noninvasive approach to map myocardial iron. It applies a strong magnetic field (~3 T) that induces an additional magnetic field (~μT) that depends on the tissue magnetic susceptibility distribution and its orientation with respect to the applied field[19]. Quantitative susceptibility mapping (QSM) is a technique that quantifies the magnetic susceptibility using the induced field and is measured directly using the MRI signal phase. Biometals such as iron perturb the MRI magnetic field and can be detected using this technique.

To investigate the role of iron in reperfusion injury, we associated QSM with coronary occlusion duration in a large animal model and in patients with successful reperfusion after ST-elevation myocardial infarction (STEMI). The feasibility of this approach was recently shown in healthy subjects[20], however, no prior studies have investigated the relationship between magnetic susceptibility and myocardial tissue iron content with myocardial infarction in animal models or patients. Magnetic susceptibility might be a specific imaging biomarker of myocardial hemorrhage, unlike conventional MRI relaxation time contrast or mapping, which is also affected in part by inflammation and fibrosis.

The purpose of this study was to measure the magnetic susceptibility of myocardial infarction and determine its association with infarct tissue iron. We hypothesized that reperfusion injury in myocardial infarctions show a paramagnetic shift of magnetic susceptibility caused by elevated tissue iron content, which was independently determined by histology and inductively coupled plasma optical emission spectrometry (ICP-OES), electron paramagnetic resonance (EPR) spectroscopy, and RNA analysis of markers for iron metabolism. To test this hypothesis, we performed a cross-sectional study in a large animal model of reperfusion injury myocardial infarction and investigated the effect of occlusion time on tissue iron and magnetic susceptibility. QSM was compared with conventional MRI tissue characterization methods ($^1$H T2* relaxation times and LGE) and correlated to infarct size and cardiac function. We also sought to determine the evolution of iron and magnetic susceptibility during wound healing. Finally, we studied patients following reperfusion for STEMI to determine the feasibility of mapping reperfusion injury using QSM.

## Results

### Animal model of myocardial reperfusion injury.
A swine model of reperfused and permanent myocardial infarction was generated by occlusion of the left anterior descending (LAD) or circumflex coronary artery branches[21–23]. To investigate the association between iron, magnetic susceptibility, and severity of infarction, coronary occlusion was performed in swine for varying duration: 45 min ($n = 3$), 90 min ($n = 5$), 180 min ($n = 5$) and permanent occlusion ($n = 4$) at 1 week after infarction. After mapping the affected territory (Supplementary Fig. 1a), animals showed discoloration of the myocardium upon visual inspection during surgical ligation (Supplementary Fig. 1b, c), hypokinetic wall activity on echocardiogram (Supplementary Fig. 1d, e), and STEMI on electrocardiogram (ECG) (Supplementary Fig. 1f–h).

At 1 week after occlusion, cine MRI showed that all animals had comparable end-diastolic and end-systolic volumes (EDV and ESV), ejection fraction (EF), and left ventricle (LV) mass (Supplementary Table 1), which suggests that animals did not show early LV remodeling or substantial myocardial hypertrophy. Late gadolinium enhanced (LGE) MRI showed that longer time-to-reperfusion was associated with larger infarct size ($P = 0.05$), greater microvascular obstruction (MVO) ($P = 0.05$) and increased infarct transmurality ($P = 0.02$), as detailed in Supplementary Fig. 2 and Supplementary Table 1. Representative LGE MRI from each group is shown in Fig. 1a.

### Association of infarct magnetic susceptibility and iron.
To determine the magnetic susceptibility of infarcted myocardium, we performed cardiac QSM of whole heart ex vivo specimens at 1 week after coronary artery occlusion. Quantitative susceptibility maps were generated from magnitude and phase gradient-echo MRI of explanted swine hearts at 0.7 mm$^3$ voxel size immediately following in vivo MRI as detailed (see "Methods" and Supplementary Fig. 3). The magnetic susceptibility of the infarct groups were $\Delta\chi_{45} = 0.03 \pm 0.05$, $\Delta\chi_{90} = 0.07 \pm 0.04$, $\Delta\chi_{180} = 0.06 \pm 0.03$, and $\Delta\chi_P = 0.03 \pm 0.01$ ppm compared with remote myocardium $0.003 \pm 0.02$ ppm. There was a significant paramagnetic shift in magnetic susceptibility in infarcted regions compared with remote myocardium in 90 min ($P = 0.012$) and permanent ($P = 0.025$) infarct groups (Fig. 1b).

To determine if magnetic susceptibility was associated with iron content, we quantified total iron using ICP-OES. Iron concentrations in infarct regions were elevated compared with remote myocardium in all groups ($[Fe]_{45} = 0.08 \pm 0.04$, $[Fe]_{90} = 0.11 \pm 0.07$, $[Fe]_{180} = 0.18 \pm 0.07$, $[Fe]_P = 0.12 \pm 0.14$ mg/g, and $[Fe]_{remote} = 0.04 \pm 0.02$ mg/g) and colorimetric assay of total and Fe$^{2+}$ iron revealed the same trend (Fig. 1c and Supplementary Fig. 8a).

Histologic evaluation was performed to investigate the spatial distribution of iron in the myocardium. Spatial correspondence between histology staining and quantitative susceptibility maps was achieved using fiducial landmarks including the RV insertion, papillary muscle shape, length of the LV, and distance to the apex. Swine reperfused after 90 min showed nontransmural infarcts with dense peripheral extracellular matrix, a central zone of nonviable cardiomyocytes, and red blood cells. There was iron accumulation at the transition zone between necrotic myocytes and mixed viable myocytes suggesting an active immune response originating outside the infarct core (Fig. 1d). Permanently infarcted myocardium showed extensive fibrosis, transmural injury, and substantially less positive iron staining (Fig. 1e) indicating a difference in wound healing and tissue remodeling. Viable myocardium did not show positive staining for iron, fibrosis, myocardial cell death, or hemoglobin (representative histology from each infarct group and remote myocardium are shown in Supplementary Figs. 6 and 7).

We further investigated the nature of elevated iron using EPR, gene expression assays and immunohistochemistry (IHC). Iron was significantly increased in the infarct region as shown by total

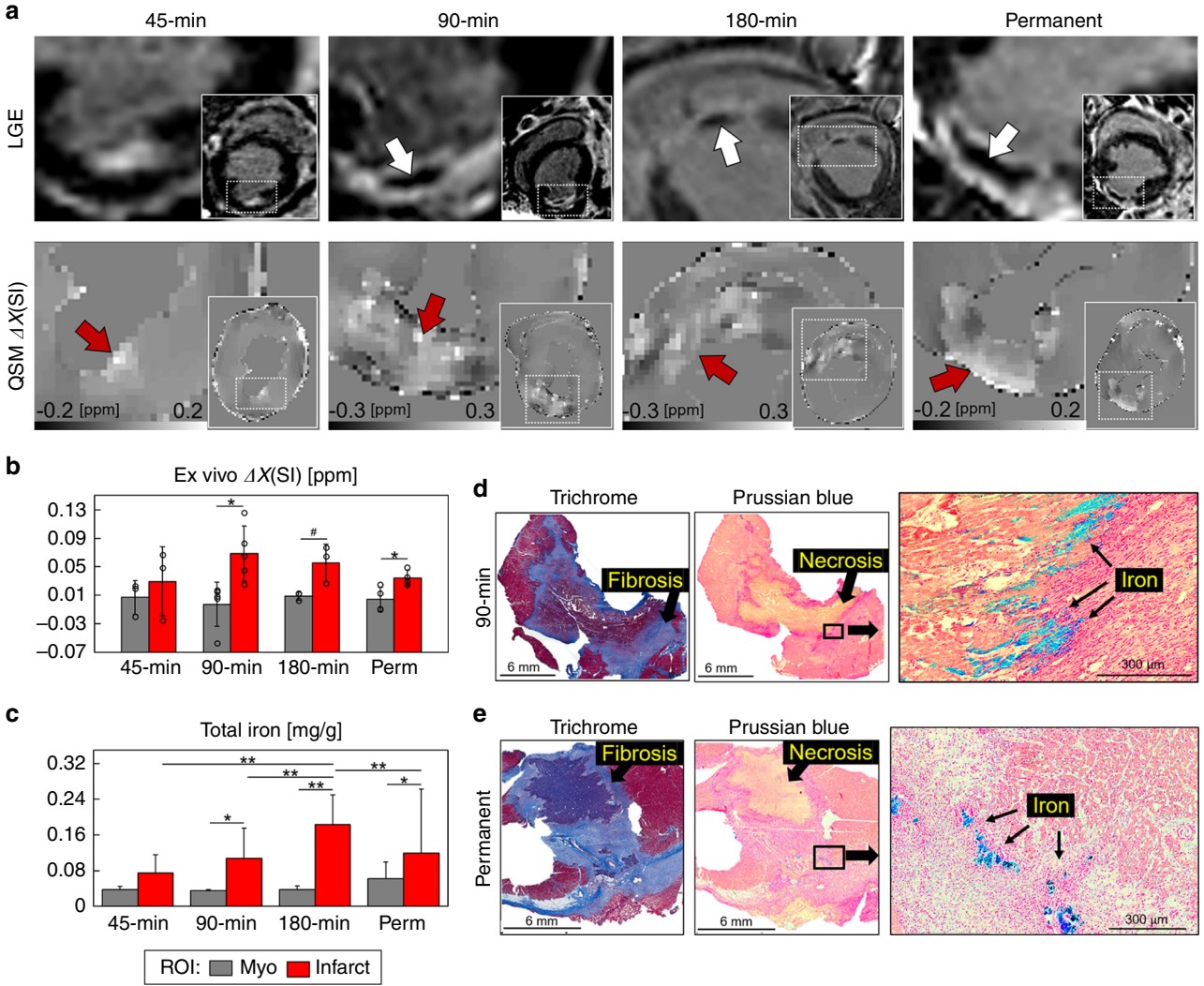

**Fig. 1 Infarct magnetic susceptibility is associated with elevated tissue iron content.** At 1 week post-infarction (**a**), late gadolinium enhanced (LGE) MRI shows hyperintense signal in the infarct region corresponding to uptake of gadolinium contrast agent. In total, 90, 180 min, and permanent infarcts were larger with microvascular obstruction, as indicated by hypointense regions in the infarct core (white arrows). Images show views of the infarct region and left ventricular short axis (insets). Infarcts show a paramagnetic shift in ex vivo quantitative susceptibility maps (QSM) for each infarct group compared with remote (myo) regions. **b** There was a significant infarct paramagnetic shift in 90 min ($P = 0.012$) and permanent ($P = 0.025$) infarct groups. Magnetic susceptibility measurements in 45 min ($n = 3$), 90 min ($n = 5$), 180 min ($n = 3$), and permanent ($n = 4$) infarct groups were from independent whole heart ex vivo specimens, regions of interest (ROI) were compared using a two-tailed two-sample $t$ test for each infarct group. **c** Infarct total iron concentration was significantly elevated in 90 min ($P = 0.001$), 180 min ($P < 0.0001$), and permanent ($P = 0.015$) infarcts compared with remote myocardium. In total, 45 min ($n = 31, 24$), 90 min ($n = 32, 32$), 180 min ($n = 48, 37$), and permanent ($n = 30, 50$) measurements were from independent tissue samples (myo, infarct), tissue regions and infarct groups were compared using two-way analysis of variance (ANOVA) with Tukey's HSD post-hoc test. **d** Representative histological findings in a 90 min reperfused infarct. Trichrome stain shows a nontransmural infarct, fibrosis, and nonviable myocytes at the core of the infarct. Prussian blue staining shows iron accumulation (black arrows) at the transition zone between necrotic myocytes and mixed viable myocytes suggesting an active immune response originating outside the infarct core. **e** Representative histological findings in a permanent occlusion infarct. Trichrome stain shows fibrosis and transmural infarct. Prussian blue shows iron deposits at the peripheral infarct fibrotic regions. Representative histology from each infarct group and remote myocardium are shown in Supplementary Fig. 6. Histology was repeated independently on multiple tissue sections ($n > 5$) for each whole heart specimen ($n = 15$) showing similar histopathological findings for each infarct group. The results are reported as mean ± SD. $^{\#}P = 0.094$ and significance $^{*}P < 0.05$ and $^{**}P < 0.001$. Source data for **b** and **c** are provided as a Source data file.

iron measurements from ICP-OES ($P = 7e-22$) and desferrioxamine-chelatable labile iron from EPR ($P = 3e-4$) (Fig. 2a and Supplementary Fig. 8b). To confirm the increase of iron, we tested the gene expression of cellular iron metabolism markers in infarct ($n = 24$) and remote myocardium ($n = 20$) tissue samples. Ferritin is a scavenger of intracellular labile iron, arresting its redox activity. *Ferritin light chain* (*FLC*) expression significantly increased in the infarct (infarct, 1137.2% Myo vs. remote myocardium, 88.1% Myo, $P = 0.002$) and *ferritin heavy*

*chain* (*FTH1*) expression was not significantly modified in infarct compared to remote area in any of the conditions studied (Fig. 2b). The expression of the intracellular iron sensor *Iron Regulatory Protein 2* (*IRP1*, also known as *ACO1*) was significantly decreased in the infarct region (infarct, 61.4% Myo vs. remote myocardium, 102.4% Myo, $P = 0.035$) (Fig. 2c). *Divalent metal transporter 1* (*DMT1*), which transports iron into the labile iron pool, was significantly decreased in the infarct region (infarct, 53.7% Myo vs. remote myocardium, 99.1% Myo,

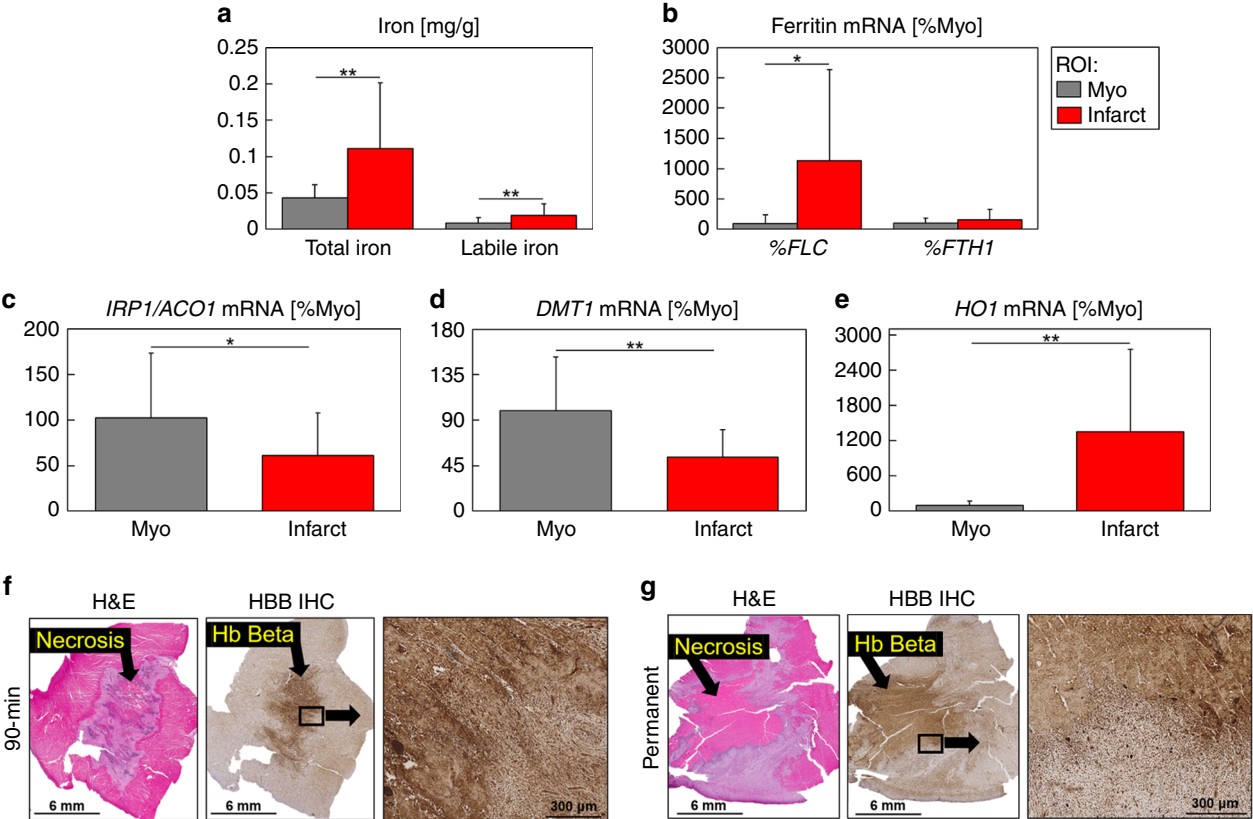

**Fig. 2 Ex vivo iron content and expression of cellular markers of iron homeostasis. a** Infarcts had significantly increased total iron ($P = 7e-22$) and labile iron ($P = 3e-4$) concentration compared with remote (myo) regions. Myo ($n = 188$) and infarct ($n = 216$) total iron ICP-OES and myo ($n = 25$) and infarct ($n = 43$) labile iron EPR measurements were from independent tissue samples. **b** *Ferritin light chain* (*FLC*) expression was significantly increased in infarct regions ($P = 0.002$) and *ferritin heavy chain* (*FTH1*) expression was not significantly modified ($P = 0.15$). **c** The expression of the intracellular iron sensor *iron regulatory protein 2* (*IRP1*, also known as *ACO1*) ($P = 0.035$), and **d** *divalent metal transporter 1* (*DMT1*), which transports iron into the labile iron pool ($P = 0.002$), were significantly decreased in infarct regions. **e** *Heme oxygenase-1* (*HO1*) expression was significantly increased in infarct regions ($P = 2e-4$). Total mRNA was isolated from myo ($n = 20$) and infarct ($n = 24$) independent frozen tissue samples and real-time PCR for quantification of mRNA was performed by using an SYBR-Green protocol. The results are expressed as fold changes in expression when compared with the average of remote samples from all time points using the cycle threshold 2($\Delta\Delta CT$) method with *GAPDH* and *HPRT* as reference genes. Total and labile iron, *FLC*, *FTH1*, *IRP1*, *DMT1*, and *HO1* measurements were compared between myo and infarct regions using a two-tailed two-sample *t* test. **f** Representative 90 min reperfused infarct at 1 week post-infarction shows a lack of cell viability (H&E) and positive hemoglobin (Hb) beta staining (HBB IHC). **g** Representative permanently occluded infarct at 1 week post-infarction shows similar dark Hb β staining, with less concentrated areas of positive staining. The darker staining may reflect extracellular or uptake of Hb β from past hemorrhage. Representative histology from each infarct group and remote myocardium are shown in Supplementary Figs. 6 and 7. Histology was repeated independently on multiple tissue sections ($n > 5$) for all reperfused and permanent infarcts which showed similar histopathological findings within each infarct group. The results are reported as mean ± SD. Significance is indicated by *$P < 0.05$ and **$P < 0.001$, region of interest (ROI). Analysis is across all reperfusion groups. Source data for **a**–**e** are provided as a Source data file.

$P = 0.002$) (Fig. 2d). *Heme oxygenase-1* (*HO1*) transcription is activated in response to increased intake of heme proteins, and its expression was significantly increased (infarct, 1300% Myo vs. remote myocardium, 100% Myo, $P = 2e-4$) (Fig. 2e). There was increased hemoglobin staining by IHC in reperfused and permanently occluded animals, compared with remote myocardium (Fig. 2f, g and Supplementary Fig. 7), which contributed to the increased *HO1* expression.

**Magnetic susceptibility imaging versus conventional MRI**. To investigate the relationship between magnetic susceptibility and conventional MRI biomarkers in ischemic myocardium, we compared LGE MRI, T2*-weighted images (T2*w), T2* maps, phase images, and susceptibility maps in Fig. 3a (additional data shown in Supplementary Figs. 4 and 5). Three regions were classified from T2*w images: (1) isointense (iso) remote myocardium, (2) hyperintense (hyper), and (3) hypointense (hypo)

infarct regions. T2*w images and T2* maps showed hypointense regions after 90 and 180 min of coronary artery occlusion but not after 45 min or in permanently infarcted myocardium. Susceptibility maps showed increased susceptibility in T2*w hypointense regions at 1 week after 90 ($P = 7e-4$) and 180 min ($P = 0.003$) of coronary artery occlusion compared with isointense regions (Fig. 3b). As expected, hypointense regions seen at 1 week after 90 and 180 min of coronary artery occlusion correlated with a decrease in T2* relaxation time and a paramagnetic shift (Fig. 3c). In contrast, the absence of both T2*w hypointensity and decrease in T2*, permanent infarcts also showed a significant paramagnetic shift in magnetic susceptibility compared with remote regions ($P = 0.025$).

Receiver operating characteristic (ROC) curves were computed for each predictor variable (magnetic susceptibility, T2* relaxation time, R2* relaxation rate) with binary response variable as infarct equal to 1 and remote myocardium equal to 0 (Supplementary Fig. 10). Magnetic susceptibility had increased

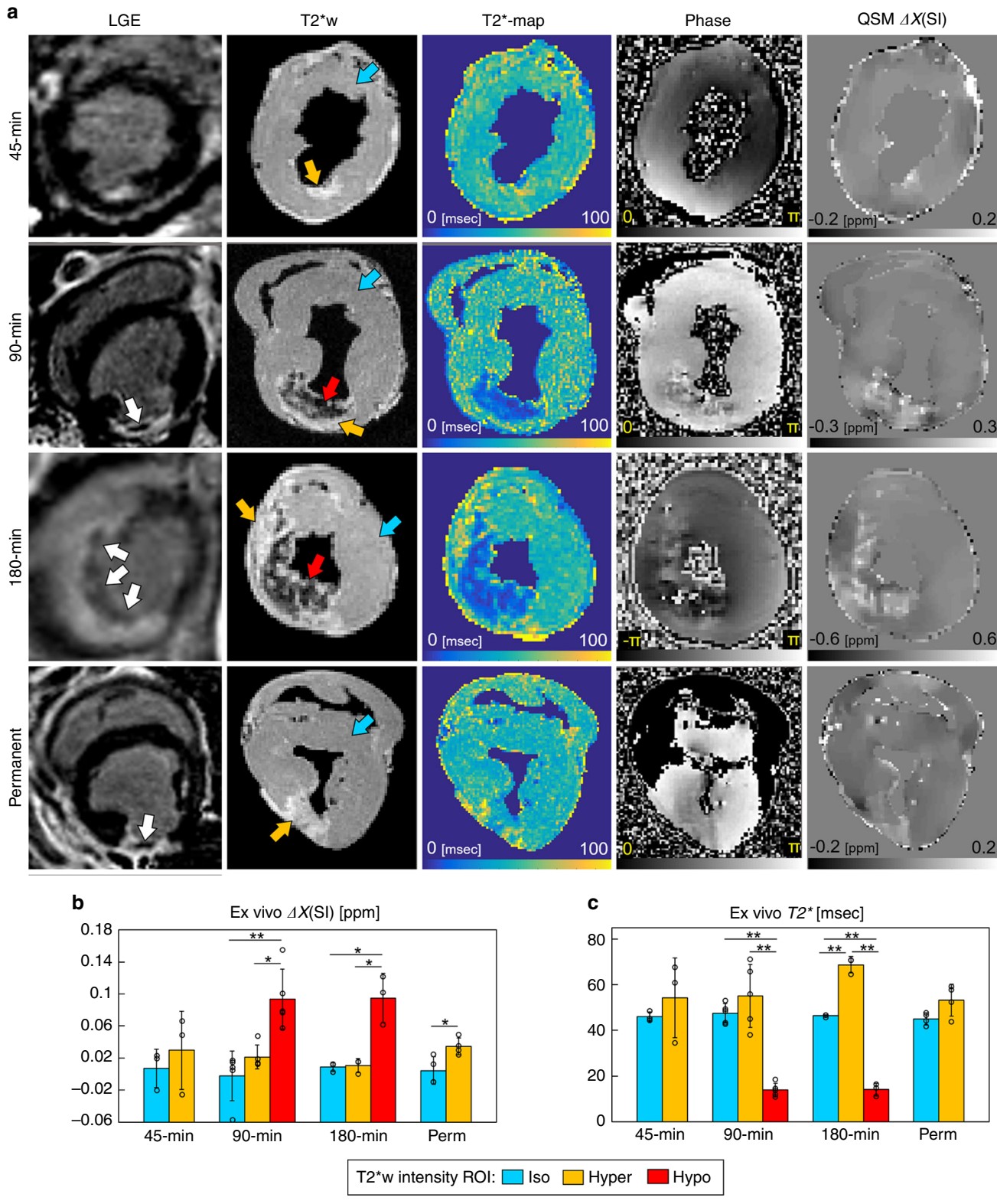

**b** Ex vivo $\Delta X$(SI) [ppm]

**c** Ex vivo $T2^*$ [msec]

T2*w intensity ROI: ▮ Iso ▮ Hyper ▮ Hypo

area under the ROC curve (AUC) and showed superior balance between sensitivity and specificity at classifying infarct versus remote myocardium compared with $T2^*$ and $R2^*$.

**Detection of an active immune response and iron**. To investigate if tissue magnetic susceptibility and iron content was associated with inflammation at 1 week after reperfusion injury, we

performed histology of ischemic tissues. Representative images of 180 min and permanent occlusion myocardial infarcts are shown in Fig. 4. All animals showed regions of iron accumulation at the transition zone between myocyte necrosis and mixed myocyte viability colocalized with MHC class II antigen-presenting cells and newly recruited macrophages. H&E stains showed tissue damage, cell viability, and an active immune response (Supplementary Fig. 6). The central core of the infarcts had autolyzed

**Fig. 3 Comparison of conventional MRI with magnetic susceptibility imaging. a** In vivo late gadolinium enhanced (LGE), and ex vivo T2*-weighted (T2*w), T2* maps, gradient-echo phase images, and quantitative susceptibility maps (QSM) obtained 1 week after 45, 90, 180 min, and permanent coronary occlusion. In distinction to the 45 min infarct group, 90, 180 min, and permanent infarcts showed a hypointense signal on LGE MRI corresponding to delayed uptake of the contrast agent (white arrows). T2*w and T2* maps show hypointense regions after 90 and 180 min of coronary occlusion, but not after 45 min and permanent coronary occlusion. This suggests elevated iron content in reperfused infarcts with longer coronary occlusion time. Gradient-echo phase images show local signal dephasing in the infarcted myocardium in all infarct groups and corresponds to elevated susceptibility seen in QSM. **b** Magnetic susceptibility from iso, hyper, and hypo regions of interest (ROI) on T2*w images. In total, 90 min reperfused infarcts showed substantially elevated magnetic susceptibility in hypo compared with hyper ($P = 0.006$) and iso ($P = 7e-4$) regions. In total, 180 min infarcts were elevated in hypo compared with hyper ($P = 0.003$) and iso ($P = 0.003$) regions, and permanent infarcts were elevated in hyper compared with iso ($P = 0.025$) regions. **c** T2* relaxation times from iso, hyper, and hypo ROI on T2*w images. Together, (**b**) and (**c**), show elevated magnetic susceptibility in all infarct groups despite increased T2* relaxation times in 45 min and permanent infarcts. This suggests that low levels of iron may be undetectable by T2* when edema and fibrosis changes are present. Magnetic susceptibility and T2* measurements in 45 min ($n = 3$), 90 min ($n = 5$), 180 min ($n = 3$), and permanent ($n = 4$) infarct groups were from independent whole heart ex vivo specimens. ROIs were compared for each infarct group using one-way ANOVA with Tukey's HSD post-hoc test. The results are reported as mean ± SD. Significance is indicated by *$P < 0.05$ and **$P < 0.001$. T2*w and phase images are at echo time (TE) 16.1 ms. Source data for **b** and **c** are provided as a Source data file.

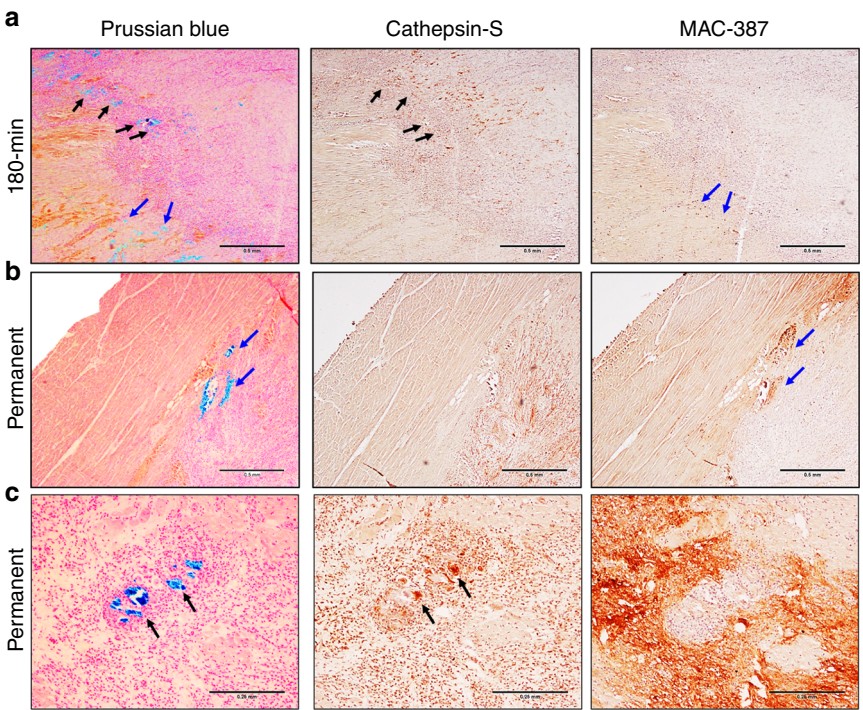

**Fig. 4 Immune response and iron deposition in reperfused and permanent infarcts. a** Representative 180 min reperfused infarct at 1 week post-infarction shows regions of iron accumulation at the transition zone between myocyte necrosis and mixed viable myocytes (Prussian blue, black, and blue arrows) colocalized with MHC class II antigen-presenting cells (Cathepsin-S, black arrows) and newly recruited macrophages (MAC-387, blue arrows). **b**, **c** Representative permanently occluded infarct at 1 week post-infarction also had heterogeneous macrophage aggregation with newly recruited macrophages (MAC-387, blue arrows) and MHC class II antigen-presenting cells (Cathepsin-S, black arrows) colocalized with iron accumulation (Prussian blue, black, and blue arrows). Histology was repeated in independent tissue sections for 90 ($n = 2$), 180 min ($n = 2$) reperfused, and permanent ($n = 2$) infarcts which showed similar histopathological findings within reperfused and permanent infarct groups. **a** and **b** are at ×40 magnification and **c** is at ×100 magnification.

myocytes with preserved cell membrane but showed pale or absent nuclei and pale cytoplasm with the loss of sarcomere cross-striations. Reperfused infarcts showed significantly more iron deposition compared with the permanent infarcts. Dense iron deposition was predominantly found in the transition zone of the infarction. Permanent infarcts had reduced erythrocyte extravasation and less iron deposition at the transition zone, although clusters of iron were found at the border of the fibrotic region of the infarct (Figs. 1e and 4b, c), indicating an active immune response with newly recruited macrophages (Fig. 4b) and iron containing MHC class II antigen-presenting cells (Fig. 4c) at the periphery of the fibrotic region. The elevation in iron concentration (Fig. 1c) along with a reduced positive

Prussian blue staining in the permanent infarct group indicates intracellular iron released from autolyzed myocytes at the core of the permanent infarct and an active immune response originating outside the infarct region.

**Magnetic susceptibility and iron evolve in wound healing.** To investigate how magnetic susceptibility and iron progress after infarction, we performed a cross-sectional MRI study in control animals without myocardial infarction ($n = 5$) and at 1 week ($n = 15$) and 8 weeks ($n = 8$) after infarction. Cardiac structural and functional data were obtained using cine and LGE MRI, and show changes in wall thickening (WT), end-systolic wall

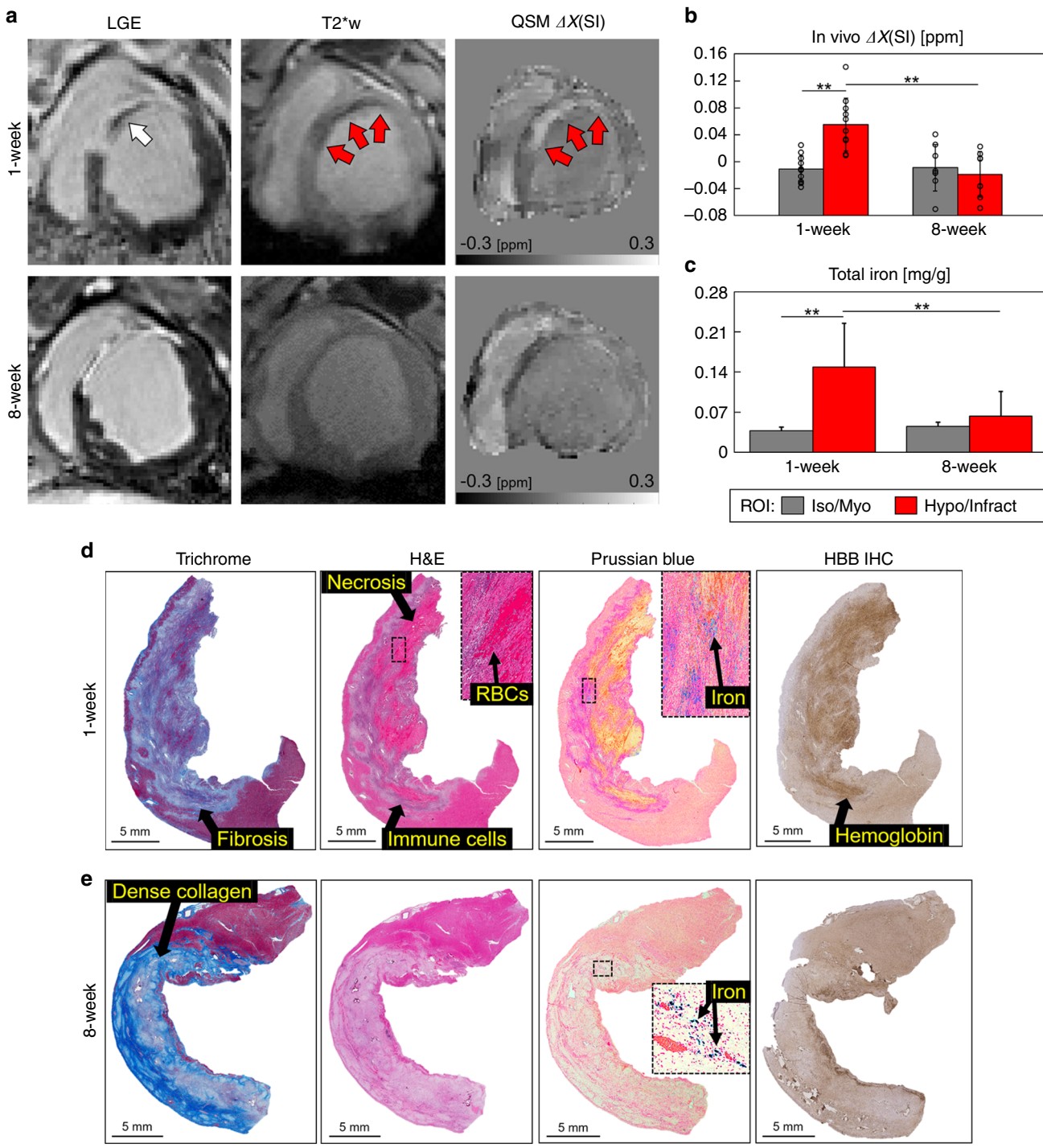

thickness, and wall motion (WM) between remote and infarct regions and between controls, 1 week and 8 week post-infarction groups (Supplementary Table 2). In vivo QSM was acquired in a subset of animals at 1 week ($n = 11$) and 8 weeks ($n = 8$) post-infarction. LGE, T2*-weighted and QSM images from one animal imaged serially at 1 and 8 weeks post-infarction are displayed in Fig. 5a (additional pigs shown in Supplementary Fig. 9). At 1 week there was a significant paramagnetic shift within the infarct region ($\Delta\chi = 0.055 \pm 0.039$ ppm) compared with the remote myocardium ($\Delta\chi = -0.011 \pm 0.020$ ppm) ($P = 2e-4$) (Fig. 5b). This was associated with elevated total iron concentration, measured by ICP-OES (infarct samples $n = 68$, [Fe] = $0.15 \pm 0.08$ mg/g vs. remote myocardium samples $n = 80$, [Fe] =

$0.04 \pm 0.01$ mg/g, $P < 0.0001$) (Fig. 5c). By 8 weeks post-infarction there was a significant decrease in both infarct magnetic susceptibility ($\Delta\chi = -0.019 \pm 0.034$ ppm, $P = 1e-4$) and total iron concentration ($n = 23$, [Fe] = $0.06 \pm 0.04$ mg/g, $P < 0.0001$). There was a significant association between MVO versus magnetic susceptibility ($R^2 = 0.31$, $P = 0.048$) and a significant association between total iron concentration and ex vivo tissue magnetic susceptibility ($R^2 = 0.48$, $P = 7e-4$) (Supplementary Fig. 11a, c).

Histology (Fig. 5d, e) showed the evolution of fibrosis and necrosis between 1 and 8 weeks post-infarction. Erythrocyte and hemoglobin deposition was seen at 1 week. Prussian blue shows iron accumulation at the transition zone between necrotic myocytes

**Fig. 5 Magnetic susceptibility and iron evolve from 1 to 8 weeks post-infarction. a** In vivo late gadolinium enhanced (LGE), T2*-weighted (T2*w), T2* maps, and quantitative susceptibility maps (QSM) from a reperfused 90 min animal imaged serially. In vivo LGE, reperfused infarcts at 1 week post-infarction had microvascular obstruction (MVO) indicated by hypointense regions (white arrows), MVO is no longer present at 8 weeks post-infarction. QSM shows an infarct paramagnetic shift (red arrows) compared with viable myocardium (myo). **b** At 1 week, reperfused infarcts had a significant paramagnetic shift ($P = 2e-4$) associated with (**c**) a significant elevation in infarct iron concentration ($P < 0.0001$). From 1 to 8 weeks post-infarction there was a significant decrease in infarct magnetic susceptibility ($P = 1e-4$) and iron concentration ($P < 0.0001$). Magnetic susceptibility was measured in vivo at 1 week ($n = 11$) and 8 weeks ($n = 8$) post-infarction from independent animals. Total iron ICP-OES measurements at 1 week ($n = 80, 68$) and 8 weeks ($n = 20, 23$) were from independent tissue samples (myo, infarct). Two-way ANOVA with Tukey's HSD post-hoc tests were used to compare across tissue regions and post-infarction time points. **d** Representative 90 min 1 week post-infarction histology shows extensive fibrosis (Trichrome), erythrocyte deposition (H&E, inset), and iron accumulation (Prussian blue, inset) at the transition zone between autolyzed necrotic myocytes (H&E) and mixed viable myocytes. Immunohistochemistry shows positive staining for hemoglobin (HBB IHC). **e** Representative 90 min 8 weeks post-infarction histology shows dense collagen deposition (Trichrome), lack of cell viability (H&E), sparse regions of iron accumulation (Prussian blue, inset) within the infarct core and there was a lack of positive hemoglobin staining (HBB IHC). Representative histology from each infarct group and remote myocardium are shown in Supplementary Figs. 6 and 7. Histology was repeated independently on multiple tissue sections ($n > 5$) for each post-infarction time point which showed similar histopathological findings within each group. The results are reported as mean ± SD. Significance is indicated by $**P < 0.001$. T2*w images are at echo time (TE) 10 ms. Source data for **b** and **c** are provided as a Source data file.

and mixed viable myocytes along with an active immune response. At 8 weeks post-infarction there was a lack of erythrocytes and immune cells. Prussian blue showed sparse iron accumulation localized at the infarct core and there was a lack of positive hemoglobin staining (representative histology from each post-infarction time point is shown in Supplementary Figs. 6 and 7).

**Magnetic susceptibility in STEMI patients.** The large animal model preclinical findings demonstrated the potential for clinical translation and formed the basis of an observational, cross-sectional MRI study in humans. Seven STEMI patients (mean ± SD, age $= 61 \pm 9$ years, one female, six males; Supplementary Table 3) treated by primary percutaneous coronary intervention (PCI) with successful reperfusion and without a previous myocardial infarction underwent cardiovascular MRI at 1.5 T. All subjects gave informed consent to participate in a research study approved by the Institutional Review Board of the University of Pennsylvania. The examinations were performed at $2.9 \pm 1.5$ days following primary reperfusion and included T2*w, T2* relaxation time mapping, cardiac QSM, and LGE MRI.

Four of seven patients had hemorrhagic myocardial infarction as determined by >1 g of hypointense region on T2* MRI. Representative LGE, T2*w, and QSM are shown in three subjects (Fig. 6, all subjects appear in Supplementary Fig. 12, reproducibility analysis in Supplementary Fig. 13, and MRI findings in Supplementary Table 4). Hemorrhage was observed at least once in the right coronary artery ($n = 2$), LAD artery ($n = 3$), and left circumflex artery ($n = 1$). All patients ($n = 3$) with TIMI grade 0/1 prior to revascularization had myocardial hemorrhage while three of four patients with TIMI grade 2/3 did not have hemorrhage. After intervention, all patients had successful reperfusion with TIMI grade 3 flow. Infarct size and regions of MVO tended to be larger in patients with hemorrhage (45.9 g [IQR = 38.6, 52.8] g) compared with those without (31.9 g [IQR = 14.0, 43.9]). The magnetic susceptibility of the infarct in the hemorrhage group ($n = 4$, $0.16 \pm 0.09$) was significantly elevated compared with the remote myocardium in all patients ($n = 7$, $-0.01 \pm 0.06$ ppm, $P = 0.005$). The duration from symptom onset to reperfusion was 303.7 min in the no hemorrhage group and 695.5 min in the hemorrhage group (Supplementary Table 3). Three of four patients who showed a magnetic susceptibility alteration had transmurality > 75% and MVO > 5% LV mass. Of the patients who we did not detect a change in magnetic susceptibility, none had transmurality > 75% and only 1 had MVO > 5%.

## Discussion

This study investigated the association between reperfusion therapy and magnetic susceptibility, an endogenous imaging biomarker of tissue iron, in a clinically relevant large animal model of hemorrhagic myocardial infarction and in patients after reperfusion therapy for acute myocardial infarction. We found that there was a paramagnetic shift in tissue magnetic susceptibility corresponding to the affected coronary vascular bed and this was associated with spatially elevated tissue iron content. Tissue magnetic susceptibility was closely associated with the duration of coronary artery occlusion; yet while animals with longer time-to-reperfusion showed significant changes in magnetic susceptibility, also animals that did not undergo reperfusion after myocardial infarction showed magnetic susceptibility changes and elevated tissue iron content in comparison to remote myocardium.

The development of methods to image myocardial ischemia reperfusion injury noninvasively is essential to improve our understanding of pathophysiology and determine risk factors for adverse LV remodeling and arrhythmias. Elevated magnetic susceptibility shown on QSM was associated with accumulation of post-infarction tissue iron, increased desferrioxamine-chelatable iron and active cellular iron homeostasis response, which were independently validated with histology and ICP-OES, EPR, and gene expression analysis by real-time PCR. The combination of susceptibility maps with LGE MRI and relaxation time mapping may substantially improve our understanding of the distribution of iron in myocardial ischemia, allowing for more robust and reliable quantification and interpretation of the results. We investigated this combined approach in a preliminary study of seven patients with STEMI and showed that some, but not all patients, had regions of elevated magnetic susceptibility that were spatially consistent with the area of infarction reported by ECG, cardiac catheterization and LGE MRI. While all patients were successfully reperfused as assessed by coronary TIMI flow grade, not all patients had hemorrhage or elevated infarct magnetic susceptibility. However, the patient's infarct size ranged from 10 to 45% of the LV. This may indicate that despite adequate coronary flow, there was still myocardial tissue ischemia without successful blood perfusion causing extensive fibrotic regions in these patients (patients 3–5 (Supplementary Fig. 12)) resembling the physiology seen with permanent occlusion infarcts.

In comparison to remote myocardium, permanently occluded animals had elevated iron content, suggesting that there was elevated tissue iron even in the absence of hemorrhagic reperfusion injury. We found that moderately elevated magnetic susceptibility

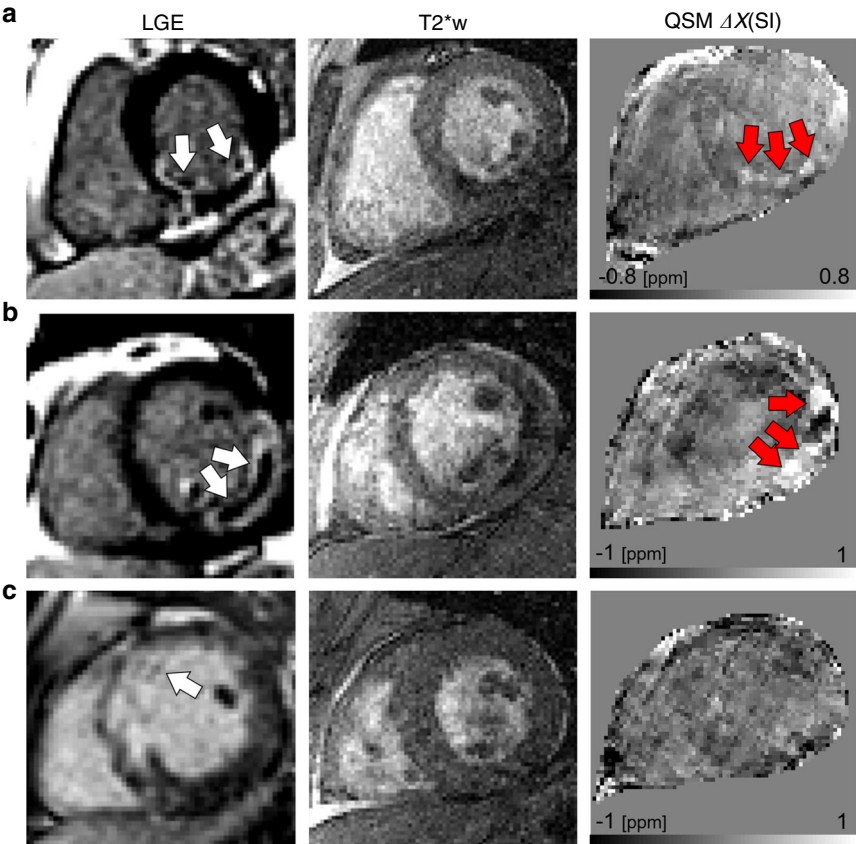

**Fig. 6 Imaging of ST-elevation myocardial infarction patients after reperfusion therapy. a** Patient had a nontransmural infarction with microvascular obstruction (LGE, white arrows), hemorrhage, and elevated magnetic susceptibility (QSM, red arrows). **b** Patient had substantial transmural infarction with microvascular obstruction, and significant hemorrhage and increases in magnetic susceptibility. **c** Patient shows a nontransmural infarction, minimal microvascular obstruction, and no substantial increase in magnetic susceptibility in the infarct region. All patients were reported to have TIMI grade 3 flow (no flow defects) following reperfusion and were imaged within 48 h after reperfusion therapy. T2*w images are at echo time (TE) 10 ms.

in permanently occluded animals did not cause substantial signal loss on T2*-weighted images or decrease in $T2*$ relaxation time on T2* maps. The reason for the discrepancy between T2* compared with total iron and magnetic susceptibility measurements is not yet fully clear and may be caused by technical factors such as field strength[24] and voxel size[25] or an incomplete understanding of the physical mechanisms underlying T2* contrast in their relation to infarct pathophysiology. There was only a moderate association between transverse relaxation rate ($R2* = 1/T2*$ [ms$^{-1}$]) and iron ($R^2 = 0.19$, $P = 0.011$), but this improved when excluding the 45 min and permanent infarct groups ($R^2 = 0.58$, $P < 0.0001$) (Supplementary Fig. 11d, e). It has been observed that there is an association between the duration of ischemia, mobilization of iron in and out of cells, and the labile iron fraction[26–28]. A difference in $R2*$ could occur in the reperfused group by an alteration in the rate of water exchange with protein-bound iron or labile iron within cells. Alternately, differences in the clustering or aggregation of iron post-infarction may result in altered relaxation times. It was recently shown in patients that the amount of hemorrhage peaks on day 2 post-infarction using T2* MRI[1], which suggests an underlying pathophysiologic change such as the accumulation of protein-bound iron in macrophages that contributes substantially more to the transverse relaxation rate $R2*$. However, technical factors such as image spatial resolution affect T2* relaxation times and must also be taken into consideration.

In comparing QSM to conventional gadolinium and relaxation time contrasts, there were several interesting findings. All animals

with permanent infarcts and 90 and 180 min reperfused infarcts had hypointense regions within the hyperintense infarct on LGE MRI, which indicates more severe infarctions and MVO. In the permanent infarct group, all animals showed MVO without hypointense regions on T2*. In reperfused animals at 90 and 180 min, T2*-weighted and T2* maps of reperfused infarcts had hypointense regions attributed to hemorrhage. However, the sensitivity, specificity and spatial distribution of conventional MRI contrasts and mapping techniques such as T1[29–32], T2[5,11,33–38], and T2*[8,39–41] and the MR signal phase (susceptibility weighted imaging)[42–44] to hemorrhage, edema, and fibrosis are currently active areas of investigation. T2 and T2* MRI typically appear hypointense with hemorrhage, but the spatial correspondence of T2 and T2* may not overlap and the relative contributions of hemorrhagic injury, edema, and fibrosis to the MR signal are unknown. T2 and T2* are increased in edema and fibrosis[35], but appear to decrease in hemorrhagic infarction. T2* is affected by imperfections in the MRI static magnetic field, background field heterogeneity at the lung-myocardium interface, signal fluctuations due to cardiac and respiratory motion, and fluctuations of the MR signal due to the intravascular blood oxygen level-dependent effect and venous oxygen saturation. Moreover, the effect of hemorrhagic tissue iron on the MR signal phase is nonlocal, affecting adjacent nonhemorrhagic tissues, and the presence and spatial extent of hemorrhage may be underestimated or overestimated and confounded by other tissue changes.

There has been some progress toward isolating the contribution of MVO or hemorrhage to worsening of heart disease after

myocardial infarction. In a study of more than 200 patients, myocardial hemorrhage was associated with adverse remodeling and hemorrhage, but not MVO, and was associated with a composite hazard outcome of cardiovascular cause of death or heart failure event post-discharge[1]. In an additional 30 patients who underwent serial imaging, MVO and hemorrhage showed a distinct time course post-infarction. Myocardial iron appears to be retained in some patients for months after the acute event and 13–15 patients had evidence of residual iron deposition using T2* imaging at 5 months after infarction even in the absence of MVO[45,46]. We found that all animals that underwent permanent coronary occlusion showed significant MVO, yet the spatial distribution of iron deposition as observed on T2* or QSM did not match MVO on LGE (Supplementary Fig. 4). In two patients we also found limited MVO and no apparent hemorrhage (patients 4 and 5, Supplementary Fig. 12). Given these findings, it is apparent that MVO and hemorrhage are distinct pathophysiologic traits that sometimes appear coincident. LGE contrast shows the accumulation of a T1 contrast agent (gadolinium) within the extracellular space of the myocardium. Pathology that limits transport of gadolinium to the myocardium, such as an obstructed artery or MVO is expected to manifest the same patterns of contrast. QSM is a nongadolinium-based contrast and is sensitive to changes in the MR signal phase. Myocardial deposition of gadolinium-based contrast agents appears to have limited influence on the MR signal phase relative to iron found in hemorrhagic MI. We found that significant phase changes were present in hemorrhagic myocardium even in the absence of gadolinium contrast agent in one animal (Supplementary Fig. 4).

Our analysis showed several aspects of iron accumulation and cellular handling in reperfusion injury, where there was an elevation in total iron concentration and positive Prussian blue staining in reperfused and permanent infarcts. In assessing the underlying biological processes of iron, we focused on intracellular and extracellular iron. During homeostasis, most of the intracellular iron is bound to ferritin and is redox inactive. A small fraction of intracellular iron belongs to the labile iron pool to be used in intracellular reactions[26]. The iron in the labile pool is redox active, and has the potential of generating free radicals upon injury[27,28]. IRP1/ACO1 acts as an intracellular sensor of iron levels: in low intracellular iron conditions it enhances *DMT1* expression to increase iron uptake and accumulation in the labile iron pool and represses ferritin expression; in high intracellular iron conditions, it inhibits *DMT1* and de-represses iron responsive elements in the ferritin genes, increasing their expression[27]. EPR analysis revealed significantly increased infarct chelatable iron compared with remote myocardium. Desferrioxamine-bound tissue iron detected by EPR likely corresponds to loosely bound iron, which has the potential of catalyzing hydroxyl radical formation. The expression of FLC was significantly increased in infarct regions, suggesting an effective cellular response toward sequestration of intracellular labile iron by increased ferritin. FLC transcription has been reported to be more responsive to high iron levels than the heavy chain *FTH1*[47], which was not significantly modified within infarcts. Unlike iron-storage tissues like the liver in which light chain-rich ferritin is more abundant, in the heart the ferritin heavy chain is the dominant subunit. A study in a mouse model of iron overload showed that while *FTH1* mRNA levels were constitutively higher in the heart than *FLC* mRNA, *FLC* expression was more sensitive to increased cardiac iron levels[48]. Increased expression of FLC in the infarct area may potentially change the composition, as well as the functional role, of cardiac ferritin following ischemia reperfusion. A significant decrease in *DMT1* in infarct regions suggests increased intracellular iron, which would induce IRP1/ACO1-mediated inhibition of *DMT1* translation to decrease iron uptake.

For extracellular iron, immunostaining of hemoglobin showed that leftover hemoglobin from hemorrhage is present at 1 week (90 and 180 min infarcts) and declines by 8 weeks after reperfusion injury, indicating a clearance of hemoglobin during wound healing. Permanent infarct groups showed similar positive hemoglobin, with less concentrated areas of positive staining at 1 week post-infarction. The darker IHC staining contrast may reflect extracellular or uptake of hemoglobin from past hemorrhage. Hemoglobin may be an important source of magnetic susceptibility and iron content of the tissue, especially in the 90 and 180 min infarct groups. Significantly increased *HO1* within infarct regions suggests cellular clearance of extracellular hemoglobin and subsequent increased denatured hemoglobin requiring heme ring degradation and iron release. Hemoglobin converts into paramagnetic protein forms such as deoxyhemoglobin and methemoglobin and then crystalizes into hemichromes, ferritin, and hemosiderin. Of these forms, deoxyhemoglobin and methemoglobin are paramagnetic (4 and 5 free electrons) and ferritin and hemosiderin are superparamagnetic (many unpaired electrons). The ionization state, $Fe^{2+}$ or $Fe^{3+}$, as well as the molecular form of protein-bound iron, is expected to affect QSM.

There are several challenges to accurately estimate of myocardial magnetic susceptibility in humans. There are errors caused by cardiac motion, respiratory motion, blood flow, and imperfections in local tissue field estimation. These cause artifacts in the magnetic susceptibility images and hinder accurate estimation. A combination of ECG gating, apnea and magnetic field gradient flow compensation helped mitigate the adverse effects of cardiac motion, respiratory motion, and blood flow, respectively. A major source of artifacts was the presence of large significant background magnetic field gradients at the myocardium–lung interface. While these artifacts were readily identified in both patients and animals, it is necessary to develop methods that limit these effects across the heart muscle. We addressed these issues of background field heterogeneity and motion in animals by embedding the whole heart tissue explants in a susceptibility-matched fluid. In addition, we showed the feasibility to perform cardiac susceptibility mapping and detect regions of altered magnetic susceptibility in patients, spatially corresponding to the areas of infarction on LGE MRI. There may be differences between the 1 week timepoint in animals compared with the subacute (3 day) period in humans that limit extrapolation of the findings. We investigated QSM in swine (n = 3) 3 days after reperfusion and found magnetic susceptibility changes were detected in both animals and humans (Supplementary Fig. 9). In addition, while many of the patients showed changes in magnetic susceptibility, some did not; this could be explained by less severe infarctions, which was represented by the 45 min animal group. Although in principle magnetic susceptibility is field independent, differences in field strength between animals (3 T) and humans (1.5 T) may result in measured differences in susceptibility, since the induced magnetic field is stronger at higher field strength, altering the sensitivity of magnetic susceptibility imaging to different MRI measurement parameters such as echo time (TE). Misregistration between slices was a potential source of artifacts found in the patient QSM images[20]. To mitigate this issue, we minimized the breath-hold time for each slice (<20 s) to limit cardiac and respiratory motion across the cardiac slice direction. Free-breathing cardiac-gated GRE acquisitions using respiration gating may have the potential to further improve image quality[49].

In conclusion, we investigated the magnetic susceptibility of acute myocardial infarction using QSM in a large animal model and showed the feasibility to detect regions of magnetic susceptibility changes in STEMI patients. These noninvasive imaging methods, specific to iron, could be used to evaluate the effectiveness of therapies for reperfusion injury. This technique may

also provide reperfusion injury diagnosis and prognostication for acute myocardial infarction patients.

## Methods

**Animal study experimental design.** All swine underwent occlusion of the LAD or circumflex coronary artery branches and were partitioned into groups: reperfusion occurred at 45 min ($n = 3$), 90 min ($n = 15$), and 180 min ($n = 5$) after initiation of occlusion or permanent occlusion ($n = 4$) without reperfusion. Swine had a terminal study 1 week post-infarction. Three additional groups were studied at 3 days ($n = 3$) and 8 weeks ($n = 8$) after a 90 or 180 min occlusion followed by reperfusion and a separate set of controls without myocardial infarction ($n = 5$). The experimental model was chosen to emulate human post-PCI ischemia reperfusion injury and remodeling post-infarction.

**Animal care.** Yorkshire swine ($N = 38$; 30–35 kg at baseline, 100% male) were studied according to the protocols approved by the Institutional Animal Care and Use Committee of University of Pennsylvania (Philadelphia, PA). During all procedures, sedation was induced with 25–30 mg/kg ketamine intramuscularly, endotracheal intubation was performed, and the animal was maintained with a mixture of isoflurane 0.5–5% and oxygen was kept at 6 L during procedure; tidal volume was manipulated based on arterial blood gas readings, generally maintained at 10–20 mL/kg (Drager anesthesia monitor, North American, Drager, Telford, PA). Anesthesia and animal temperature were monitored during surgical and imaging procedures to maintain a constant physiologic state. Arterial access was obtained at the carotid or femoral artery for measurement of intraventricular pressure (Millar Instruments, Houston TX). Venous access was obtained at the internal jugular veins for administration of medication. After each procedure, the animal was weaned from anesthesia and transported to the recovery room. Upon completion of the terminal MRI study, the animal was returned to the operating room for euthanasia and cardiac tissue harvest.

**Animal cardiovascular surgical procedure.** A prophylactic antiarrhythmic regimen of 150 mg amiodarone, 1 mg/kg lidocaine, and 1 g magnesium sulfate were administered intravenously. A left thoracotomy was performed, and the pericardium was opened. One or more coronary snares were positioned at the branches of the circumflex artery ($n = 4$) or LAD artery ($n = 34$) to induce an infarction of ~20–30% of the LV size[21,22]. Swine underwent open chest surgery ($n = 17$) or balloon catheter ligation ($n = 16$). For open chest surgery, the dimensions of the infarction were determined by visible color changes at the epicardium (Supplementary Fig. 1b, c). The exact locations of the ligation sutures (Supplementary Fig. 1a) were decided upon after gross inspection of arterial anatomy, unique to the animal. Ischemia was confirmed by ST segment elevations on ECG (Supplementary Fig. 1f–h). After 45, 90, or 180 min of coronary artery occlusion, the coronary snare was removed, an inter-costal nerve block was performed with bupivacaine at the surgical site, and the chest was closed in muscle layers. For permanent occlusions (nonreperfused), the coronary snare was not removed. For balloon catheter occlusion, a 6Fr Hockey-stick guide catheter (Cordis Corporation, Fremont, CA) was positioned in the left main ostium under fluoroscopic guidance. A 0.18″, 180 cm Choice-PT angioplasty wire (Boston Scientific, Marlborough, MA) was carefully advanced into the LAD coronary artery. A rapid-exchange 2.5 × 12 mm angioplasty balloon (Apex, Boston Scientific, Marlborough, MA) was placed over the angioplasty wire into the mid-LAD. Serial coronary angiography was performed to position the angioplasty balloon immediately distal to the second diagonal branch of the LAD. The angioplasty balloon was inflated and maintained at 10–12 atm throughout the infarct procedure. After initial balloon inflation, repeat coronary angiography was performed to confirm adequate distal occlusion of the LAD. Uninterrupted coronary occlusion was maintained for 90 or 180 min with confirmation of acute MI by ST segment elevation in the precordial ECG leads[23].

**Animal cardiovascular magnetic resonance protocol.** MRI studies were performed on a 3 T whole-body system (Trio; Siemens Healthcare, Erlangen, Germany) with 40 mT/m gradients and 18 channel RF receiver arrays. Intraventricular pressure was interfaced to physiological monitoring software and filtered to facilitate dual respiratory and cardiac gating (LabView, National Instruments, Inc., Austin Texas). All 2D images were acquired in the short axis during apnea and 3D with dual cardiac and respiratory gating. Apnea was achieved by temporarily disabling the animal ventilator.

In vivo retrospective, short axis, multi-slice cine MRI was performed with a TR = 41 ms, TE = 2.84 ms, FOV = 34 × 20 cm², spatial resolution = 1.3 × 1.3 mm², slice thickness = 6 mm, number of slices = 18–20, flip angle = 12°, bandwidth = 399 Hz/pixel, parallel imaging acceleration factor = 2, NEX = 1 and 3–4 slices with 60 cardiac phases per slice were acquired during a breath hold of 20 s followed by a 1 min break between each breath hold. The animals received a 0.1 mmol/kg intravenous injection of gadolinium contrast for LGE imaging (MultiHance; Bracco Diagnostics, Inc.; Princeton, NJ). Imaging was performed 10 min after injection of contrast agent using a inversion time (TI) scout sequence to determine the optimal TI to null normal myocardial tissue signal. LGE MRI was obtained using a cardiac and respiratory gated 3D multishot phase-sensitive inversion recovery (PSIR)

bSSFP sequence with a TR = 443 ms, TE = 2.42 ms, FOV = 35 × 219 cm², voxel size = 1.4 × 1.4 × 4 mm³, number of slices = 26, flip angle = 20°, bandwidth = 244 Hz/pixel, parallel imaging acceleration factor = 2, NEX = 2, and total acquisition time ranged from 7 to 9 min. Respiratory and cardiac-gated 2D T2*-weighted single-echo gradient-echo images were obtained using TR = 160 ms, TE = 10 ms, FOV = 24.0 × 17.3 cm², spatial resolution = 0.9 × 0.9 mm², slice thickness = 4 mm, number of slices = 10–15, flip angle = 13°, bandwidth = 260 Hz/pixel, parallel imaging acceleration factor = 2, NEX = 12, and each slice was acquired during a breath hold of 60 s followed by a 3 min break between each breath hold.

Ex vivo imaging at 3 days, 1 week and 8 weeks post-infarction, animals were euthanized, their hearts were excised and bathed in non-$^1$H magnetic susceptibility-matched fluid (Fomblin), and T2*-weighted multiecho gradient-echo images of whole heart specimens were obtained using TR = 42 ms, 5 TEs = 3.3–16.1 with ΔTE = 3.2 ms, FOV = 23 × 23 cm², voxel size = 0.9 × 0.9 × 0.9 mm³, flip angle = 16°, bandwidth = 610 Hz/pixel, NEX = 1, and total acquisition time was 30 min.

**Eligibility criteria in humans.** Patients ($n = 7$, age = 61 ± 9 years, 1 female, 6 male) enrolled in this study had ST-elevation acute myocardial infarctions and were treated by primary PCI. The trial was conducted in accordance with the Declaration of Helsinki and the Guidelines for Good Clinical Practice. The trial protocol was approved by the Institutional Review Board of the University of Pennsylvania (Philadelphia, PA, USA). The full trial protocol can be accessed through ClinicalTrials.gov with trial registration number, NCT03531151. Patients were recruited and imaged at the Penn Presbyterian Medical Center during the time period from March to June 2018. Eligibility criteria included: (1) between the ages of 18 and 80, (2) must be able to read and understand English and sign the informed consent form, (3) elevated and delayed peak creatine kinase-MB and troponin I (cTnI) and troponin T (cTnT) in blood serum, (4) no contraindications to MRI such as claustrophobia, (5) adequate renal function, as determined by glomerular filtration rate > 30 mL/min, (6) no presence of cardiac pacemaker or implanted cardioverter defibrillator, (7) were not pregnant, (8) did not have a personal or family history of hypertrophic cardiomyopathy, (9) did not have a history of seizure disorder. The indications for PCI were based on ECG criteria showing 1 mm ST elevation in precordial leads or 0.5 mm ST elevation in limb leads in at least 2 contiguous leads. A minority of patients received thrombolytic therapy. The indications for PCI in these cases were as follows: (1) lack of resolution of ECG changes, as determined by >50% resolution of ST elevation from maximum elevation, (2) significant resolution of chest pain, as assessed by verbal confirmation with the patients, and (3) hemodynamic instability, as determined by noninvasive blood pressure measurement, heart rate changes on ECG and clear lungs.

**Human cardiovascular magnetic resonance protocol.** MRI studies were performed on a 1.5 T whole-body system (Aera; Siemens Healthcare, Erlangen, Germany) with 40 mT/m gradients and 48 channel RF receiver arrays. ECGs were obtained during MRI and used for gating using a wireless ECG monitor. All 2D images were acquired in the short axis during breath-holding and 3D with dual cardiac and respiratory gating. All patients were in a head first supine position.

Retrospective, short axis, cardiac-gated, breath held, multi-slice cine MRI was performed with a TR = 33.4 ms, TE = 1.09 ms, FOV = 30 × 40 cm², spatial resolution = 2.1 × 2.1 mm², slice thickness = 8 mm, number of slices = 10–15, flip angle = 55°, bandwidth = 930 Hz/pixel, parallel imaging acceleration factor = 2, NEX = 1 and 3–4 slices with 30 cardiac phases per slice were acquired during breath hold of 10 s followed by a 45 s break between each breath hold. LGE imaging was performed 10 min after intravenous injection of 0.15 mmol/kg gadolinium contrast (Dotarem; Guerbet USA, LLC; Princeton, NJ). Imaging was performed 10 min after injection of contrast agent using an TI scout sequence to determine the TI to null myocardial tissue signal. LGE MRI was obtained using a 3D multishot PSIR turboflash sequence with navigator respiratory gating with TR = 882 ms, TE = 1.1 ms, FOV = 24 × 32 cm², voxel size = 2 × 2 × 8 mm³, number of slices = 12–15, flip angle = 40°, bandwidth = 1080 Hz/pixel, NEX = 1, and total acquisition time ranged from 5 to 6 min. Breath held cardiac-gated 2D T2*-weighted single-echo gradient-echo images were obtained using TR = 192 ms, TE = 15 ms, FOV = 23 × 30 cm², spatial resolution = 1.6 × 1.6 mm², slice thickness = 3 mm, number of slices = 10–15, flip angle = 25°, bandwidth = 400 Hz/pixel, parallel imaging acceleration factor = 2, NEX = 1, each slice was acquired during breath hold of 12–18 s followed by a 1 min break between each breath hold. All patients were in the MRI for a maximum duration of 1 h to acquire all protocol sequences.

**QSM image analysis.** Remote myocardium was defined as the regions showing no hyperintensity or hypointensity (isointensity) on T2*-weighted images. The ROIs for healthy myocardium (isointense), infarct (hypointense), and infarct (hyperintense) were obtained by segmenting the T2*-weighted images acquired at the last echo (TE = 16.1 ms) using thresholding active contour segmentation (ITK-SNAP 3.4, University of Pennsylvania)[50]. Ex vivo infarct ROI measurements (Fig. 1b, Supplementary Figs. 10 and 11) included both hypointense and hyperintense regions. In vivo infarct ROI measurements (Fig. 5b, Supplementary Fig. 11a) included hypointense regions. Statistical analysis was performed from whole heart (multi-slice) ROI basis for each animal.

**MRI parametric mapping**. T2* mapping was performed in Matlab-R2017B (MathWorks, Natick, MA, USA) using a two-parameter model $S = Ae^{-\frac{TE}{T2^*}}$, where $A$ is the amplitude at TE = 0. The T2* GRE magnitude images had signal decay across five TEs. The two parameters ($A$ and $T2^*$) were estimated using least-squares minimization and error maps were computed to validate the accuracy of the fit.

The principle of QSM is to estimate tissue magnetic susceptibility from the measured MR gradient-echo signal phase. The magnetic susceptibility ($\Delta\chi$) is a material property that determines how effectively it is magnetized in an applied magnetic field. $\Delta\chi$ represents the magnetic susceptibility difference from water (in parts-per-million ppm of the main magnetic field, SI units). In the forward model, the relationship between $\Delta\chi$ and the measured local tissue magnetic field ($\Phi$) is $\mathbf{F}\Phi = \mathbf{DF}\Delta\chi$, where $\mathbf{F}$ is the discrete Fourier transform operator and $\mathbf{D} = 1/3 - k_z^2/k^2$ is the dipole kernel in the Fourier domain[51]. The estimation of the local $\Delta\chi$ is inherently ill-posed due to the presence of zeros in the Fourier domain representation of the dipole kernel[52]. Hence, estimation of the susceptibility map using a direct inversion of the dipole kernel leads to artifacts. A common approach for removing the artifacts from the reconstructed susceptibility maps is the use of Bayesian regularization, which consists of a combination of least-squares data fidelity term and a data regularizing term, leveraging sparsity of the image in a suitable transformation domain to remove artifacts. Tikhonov and the total variation (TV)[53] constraints are well known data regularizers that penalize the gradient of the image to enforce a piece-wise-constant/smooth model[51,54]. Here, the morphology enabled dipole inversion technique[54–57] was used to reconstruct the magnetic susceptibility using an unconstrained formulation:

$$\Delta\chi = \min_{\Delta\chi} \lambda \|\mathbf{MG}\Delta\chi_1\| + \|\mathbf{W}(\mathbf{F}^{-1}\mathbf{DF}\Delta\chi - \Phi)\|_2^2, \tag{1}$$

where $\| \;\|_1$ is the L$_1$ norm, $\| \;\|_2$ is the L$_2$ norm, $\mathbf{M}$ is an edge map estimated from the magnitude image, $\mathbf{G}$ is a spatial gradient operator, $\mathbf{W}$ is an estimate of the signal-to-noise ratio at each voxel from the magnitude images, and $\lambda$ is a Lagrangian weight, balancing data fidelity and TV-based data regularization. The cost functional in Eq. (1) was minimized using a conjugate gradient-based implementation.

QSM images are generated from multiecho gradient-echo images as shown in Supplementary Fig. 3. Phase images have signal aliasing and are unwrapped using the algorithm, simultaneous phase unwrapping and removal of chemical shift (SPURS) with graph cuts[58]. The total field map is estimated using a linear least-squares fit to the signal phase, mask is generated by thresholding a T2* magnitude image, the tissue field map is extracted using projection onto dipole fields[59], and QSM images are estimated from the local field using Eq. (1).

**Cardiac function**. Cine MRI image series was used to calculate LV mass (Mass), EDV, ESV, EF cardiac output, WT, WM, and wall thickness at end systole. Epi- and endocardial contours were drawn manually at ED and ES using standard techniques (Qmass 7.5, Medis, Leiden, The Netherlands).

**Infarct size determination**. Infarct volume and transmurality were assessed using full width at half maximum (FWHM) thresholding with visible enhancement on LGE MRI (Qmass 7.5, Medis, Leiden, The Netherlands). After defining the infarct region with FWHM, MVO regions were defined as a hypointense core within a hyperintense region on LGE and were manually segmented. Infarct volume included hyperintense regions in LGE, selected based on FWHM and hypointense MVO regions that were manually segmented.

**Isolation of tissue samples and histology**. The excised heart from each animal was sliced (2 mm thickness) using a 330 M 13″ Prosciutto Meat Slicer (Berkel; Troy, OH). Infarct regions were sectioned into $2 \times 2$ cm$^2$ samples from short-axis cardiac slices. Similarly, from each swine cardiac specimen, at least four samples of viable myocardium were isolated from the tissue opposite of the infarct region. Contiguous tissue samples were collected for both histology, ICP-OES iron measurements, EPR spectroscopy, and gene expression analysis.

For histology, 5 μm sections were obtained from representative samples ($n = 2–10$) from each of the infarct and remote regions from every animal. These sections were stained with H&E for cell viability, Masson Trichrome for fibrosis, and Prussian Blue for iron and sections were digitally imaged at ×40 magnifications and viewed with Aperio ImageScope [v12.3.2.8013] (Leica Biosystems, Wetzlar, Germany) software. Color adjustment of Prussian blue digital images was performed (parameters: sharpness 25%, brightness −38%, contrast 85%, saturation 400%, temperature 6500–6700). IHC was done using the Dako Envision+ Peroxidase Based System (Agilent Technologies; Santa Clara, CA). Hemoglobin, MHC class II antigen-presenting cells and newly recruited macrophages were detected using a Hemoglobin Beta antibody (LSBio: LS-C755633, 1:100 dilution), Cathepsin-S antibody (GeneTex: GTX114350) and a MAC-387 antibody (Abcam: ab22506), respectively. Tissue fiducial markers, RV insertion, papillary muscle shape, length of the LV, and distance to the apex were used to correlate MR images with histology.

**Inductively coupled plasma optical emission spectrometry**. The total amount of iron deposited within each myocardial sample ([Fe] in milligrams per gram of tissue) was measured using ICP-OES. Stored samples were thawed from −80 °C, blotted, weighed, and placed in individual borosilicate glass centrifuge tubes (VWR International, Radnor, Pennsylvania, USA) with PTFE-lined caps (MedSupply Partners, LLC, Atlanta, GA, USA). Sample weights ranged from 50 to 300 mg. Seventy-five microliter of nitric acid (Fisher Scientific, Hampton, NH, USA) and 225 μL of hydrochloric acid (Fisher Scientific, Hampton, NH, USA) were added to the samples. The samples were then placed in a water bath overnight at a temperature of 50 °C. The digested samples were cooled down to room temperature, diluted with ultrapure water to a volume of 6 mL, and then filtered through 0.2 μm Nalgene syringe-filters (Thermo-Fisher Scientific, Waltham, MA, USA) and the filtrates were collected in individual 15 mL metal-free polypropylene tubes (Thermo-Fisher Scientific, Waltham, MA, USA). A set of standards with concentrations ranging from 0 to 100 ppm was prepared using a 1000 ppm iron stock solution (Inorganic Ventures, Christiansburg, VA, USA).

All samples were analyzed using a Genesis ICP-OES (Spectro Analytical Instruments GMBH; Kleve, Germany). Sample selection occurred automatically via an ASX-260 autosampler (CETAC, Omaha, NE, USA); sample introduction occurred via peristaltic pump at a rate of 1.5 mL/min through a modified Lichte nebulizer with a cyclonic spray chamber. Data was acquired using the dedicated SmartAnalyzer Vision software. Fe content measured within each sample at wavelength 259.9 nm was averaged between three duplicates and expressed as mg of Fe per L of sample by using the weight of the tissue sample and volume dilution the concentration was calculated to be mg of Fe per g of tissue.

ICP-OES was performed on multiple contiguous tissue samples of viable myocardium and infarct regions; measurements were averaged from each region within 90, 45, 180 min, and permanent ligation animal models. In all ($n = 327$) total iron concentration measurements were analyzed, at 1 week post-infarction: remote ($n = 141$) tissue samples, 45 min ($n = 24$), 90 min ($n = 32$), 180 min ($n = 37$), permanent ($n = 50$) infarct tissue samples, at 8 weeks post-infarction: remote ($n = 20$) tissue samples and 90 min ($n = 23$) infarct tissue samples.

**Electron paramagnetic resonance spectroscopy**. EPR spectroscopy was performed with a modified method based on the study by Yegorov et al.[60]. Frozen samples (30–90 mg) were weighed and homogenized in 500 μL of sterile saline with a TissueRuptor (Thermo-Fisher). Aliquots were saved for colorimetric iron measurements (Sigma MAK025). Decreasing concentrations of a stock solution of 10 mM FeSO4 in 0.2 mM HCl (100–12.5 μM) were diluted in saline to be used as standards. Two hundred microliter of each homogenate and FeSO$_4$ standards were treated for 16 h with 200 μL of 10 mM desferrioxamine (Sigma). Desferrioxamine oxidizes Fe$^{2+}$ to Fe$^{3+}$, hence the desferrioxamine-bound iron detected by EPR corresponds to both original Fe$^{3+}$ in the homogenate and Fe$^{2+}$ converted into Fe$^{3+}$. For EPR measurements, samples, standards and saline as blank (80 μL) were placed in 1/8″ teflon tubes (Cole-Parmer) and frozen in liquid nitrogen (77 K). EPR spectra of samples, standards, and saline in liquid N$_2$ were recorded using a Bruker EMX spectrometer under the following conditions: microwave frequency 9.47 GHz; power, 20 mW; modulation amplitude, 1.0 mT. EPR signal was measured at g ~4.3 to estimate desferrioxamine-iron concentration. Fe content measured within each sample was averaged between at least two duplicates and expressed as mg per g of tissue. EPR was performed on multiple contiguous tissue samples of viable myocardium and infarct regions. Representative measurements from each infarct group were averaged together, in all remote ($n = 25$) and infarct ($n = 43$) tissue samples were analyzed. The EPR technique is not without error in tissue, as the isolation may mobilize or dilute catalytic iron. All tissue was processed under the same conditions, making any mobilization, or dilution of labile iron during the isolation process comparable in all samples. However, it is possible that the actual concentration (mg/g) of labile iron in the original tissue differs from the values of our results.

**Gene expression of iron handling markers**. Total mRNA was isolated from frozen samples (remote, $n = 20$, infarct, $n = 24$) using the RNeasy mini kit (Qiagen) and a TissueRuptor (Thermo). Total RNA concentration and purity were measured on a Denovix spectrophotometer (Denovix). cDNA synthesis was performed using 100 ng of RNA with Maxima H Minus First Strand Kit (Thermo Scientific). Real-time PCR for quantification of mRNA was performed on a Piko Real 96 (Thermo Scientific) by using an SYBR-Green protocol. The results were expressed as fold changes in expression when compared with the average of remote samples from all time points using the cycle threshold 2(ΔΔCT) method with GAPDH and HPRT as reference genes. Primers producing a PCR product smaller than 200 bp were designed using NIH Primer Blast with spanning an exon-exon junction when possible. The corresponding sequences were as follows: porcine GAPDH forward, 5′-ccctgcgctctctgctc-3′; porcine GAPDH reverse, 5′-gccagagttaa aagcagccc-3′; porcine HPRT forward, 5′-agccccagcgtcgtgatta-3′; porcine HPRT reverse, 5′-acatctcgagcaagccgttc-3′; porcine FLC forward, 5′-ctcatggctggtcggcaat a-3′; porcine FLC reverse 5′-tgttttggacggaacagaccc-3′; porcine FTH1 forward, 5′-ttg ggtctgcagcttcatca-3′; porcine FTH1 reverse, 5′-gccagaactaccaccaggac-3′; porcine DMT1 forward, 5′-ccaggatctagggcatgtgg-3′; porcine DMT1 reverse, 5′-actggtggct tcttcagtcag-3′; porcine ACO1 forward, 5′-accattgccaacatgtgtcc-3′, porcine ACO1

reverse, 5′-agtctgggtcttgagaggagt-3′, porcine *HO1* forward, 5′-gtaccgctcccgaatgaac a-3′, and porcine *HO1* reverse, 5′-tggtccttagtgtcctgggt-3′.

**Statistical analysis**. Statistics were performed in Matlab-R2017B (MathWorks, Natick, MA, USA) and R-3.6.3 (Vienna Austria). The descriptive results are reported as mean ± SD or interquartile range. Comparisons between two groups were performed using a two-tailed two-sample $t$ test. Comparisons between more than two groups were performed using a one-way analysis of variance (ANOVA) with Tukey's HSD post-hoc multiple comparisons test or a nonparametric Kruskal–Wallis test when appropriate. Comparisons between multiple groups and two factors were performed using a two-way ANOVA with Tukey's HSD post-hoc multiple comparisons test. ROC curves were computed for each predictor variable with binary response variable as infarct equal to 1 and remote myocardium equal to 0. The data were fit to a logistic regression model. AUC and Youden's index were computed. Sensitivity and specificity were measured at the optimal operating point. Linear regression was performed to determine if there was a significant correlation between two measures and the $P$ value was generated by testing whether the β coefficient was zero using a Wald test. The results were found to be significant for $P < 0.05$. *$P < 0.05$, and **$P < 0.001$.

**Reporting summary**. Further information on research design is available in the Nature Research Reporting Summary linked to this article.

## Data availability

All datasets are available upon reasonable request to the authors and approved through the University of Pennsylvania. The source data underlying Figs. 1b, c, 2a–e, 3b, c, 5b, c, Supplementary Figs. 2, 8a, 10, 11a–e, 13, and Supplementary Tables 1, 2 are provided as a source data file. There may be restrictions on private health data. Source data from Supplementary Tables 3 and 4 were removed to protect patient confidentiality. MRI performed in the context of this research study did not influence the treatment of the patients enrolled in the study. Source data are provided with this paper.

## Code availability

The codes used in this study are publicly available at the following links: Cornell MRI Research lab, Cornell University, Quantitative Susceptibility Mapping toolbox (http://pre. weill.cornell.edu/mri/pages/qsm.html); source code—Penn Image Computing and Science Laboratory, University of Pennsylvania, ITK-SNAP 3.4 (http://www.itksnap.org/pmwiki/pmwiki.php?n=Downloads.SNAP3); source code—Medis medical imaging systems, Leiden, The Netherlands, Qmass 7.5 (https://www.medis.nl/products/qmass); custom code—Advanced Cardiovascular Imaging Lab, University of Pennsylvania, T2* mapping (git clone https://moonbri@bitbucket.org/moonbri/penn_corlab_public_repository.git). Source data are provided with this paper.

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

## Acknowledgements

We greatly appreciate support from David R. Vann, Ph.D. for training and assistance with ICP-OES in the Earth and Environmental Science Instrumentation Facility, the Pathology Core Laboratory at the Research Institute at Children's Hospital of Philadelphia for histology staining, Loewe Kasprenski and Christine Vaspoli for coordination of large animal studies, Karen Maslowski and Connie Klahn at Penn Presbyterian Medical Center and Mary Spencer and Kathleen Thomas at the University of Pennsylvania Department of Radiology for recruitment and data coordination. The authors would like to thank the National Institutes of Health National Heart Lung and Blood Institute for their support of this project R00 HL108157, R01-HL137984, R01 HL122805, R01 HL131872, Ruth L. Kirchenstein National Research Service Award (NRSA) in interdisciplinary Cardiovascular Biology (T32-HL007954), and the HHMI-NIBIB Interfaces Program (T32-EB009384).

## Author contributions

W.R.W., J.H.G., R.C.G. and G.F. obtained the funding. W.R.W., J.H.G., R.C.G., F.W.W., and B.F.M. were responsible for the concept and design of the swine study. W.R.W. and B.F.M. drafted the paper, which was revised and approved by all authors. A.I., K.O., Y.S., and C.T. performed cardiovascular surgeries of all swine models. W.R.W., J.J.P. and B.F.M. acquired swine MR images. B.F.M., M.P.S., A.T.H. and R.K. processed and analyzed swine cardiac function and scar size data. B.F.M., M.P.S., A.T.H., R.K., and N.J.J. performed the gross tissue swine sectioning. W.R.W., S.K.I., and B.F.M. implemented Matlab algorithms to compute QSM and T2* maps. G.F., E.C., and B.F.M. designed experiments for ex vivo iron concentration measurements. A.T. and E.M.H. provided equipment and laboratory training of ICP-OES experiments. B.F.M. and E.M.H. performed ICP-OES experiments and analysis. S.J. and E.C. performed EPR experiments and analysis. E.C. performed colorimetric iron assay, RNA analysis, hemoglobin IHC and drafted manuscript sections pertaining to each experiment, including EPR. S.J.K. is responsible for Cathepsin-S and MAC-387 IHC. W.R.W., W.M., and H.L. are responsible for the design of the clinical study. W.M. was responsible for identifying and providing patients. H.L. and A.G. are responsible for patient CMR analysis. B.F.M. computed patient QSM. W.R.W, B.F.M, H.S. and E.H. were responsible for statistical analysis.

## Competing interests

The authors declare no competing interests.
