## [Peer Review File · Nature Communications]

Reviewers' comments:

Reviewer #1 (Remarks to the Author):

This is a highly interesting and novel paper demonstrating utility of cardiac QSM for assessment of intramyocardial hemorrhage (IMH) in the acute myocardial infarction (AMI) setting: Intramyocardial hemorrhage is a known adverse prognostic marker, which confers risk for adverse left ventricular (LV) remodeling. Conventional cardiac magnetic resonance (CMR) approaches employ T2/T2* to assess IMH, an approach which can be imprecise. Quantitative susceptibility mapping (QSM) provides a novel approach for measuring paramagnetic field shifts induced by tissue-based iron content; this approach has not widely been used for cardiac imaging. To test utility of cardiac QSM for post-AMI IMH, the authors performed analysis in a swine model (swine) of AMI (n=17) in which QSM was performed ex-vivo (following sacrifice), as well as a pilot cohort of AMI patients (n=7). Large animal data demonstrated an association between cardiac QSM and histopathology evidenced iron content, including abnormal QSM despite negative T2* in animals with permanent coronary ligation. Human results (in post-AMI patients) similarly demonstrated an association between altered QSM and AMI related IMH, as evidenced by elevated QSM in patients with IMH using the current clinical standard of T2* (0.16 ± 0.09 vs. -0.01 ± 0.06 , $p < 0.05$). This is a well written study, for which findings are of substantial novelty and clinical relevance. Findings provide initial proof of concept for cardiac QSM as an index of post-AMI hemorrhage, setting the foundations for future research testing clinical utility of this approach.

The study could be strengthened by addressing the following major issues:

- A key limitation concerns the fact that QSM in the large animal model was performed post-mortem (ex vivo), meaning that sacrifice itself could have induced changes in myocardial tissue properties (e.g. infarct expansion). This differs from the human (clinical) protocol, and should be addressed as a source of bias and potential challenge with respect to co-localization. It should also be noted in large animal model, cardiac QSM was performed 1 week after occlusion: Rationale for this time point is unclear; lack of serial (in vivo) imaging in large animal model limits insight into whether IMH would have been larger had animals been imaged earlier after AMI, or whether temporal time course of QSM change parallels LV remodeling patterns (i.e. LV dilation and/or wall thinning)
- Animal model presents global indices of LV function (e.g. EF, volumes): Data could be further strengthened by including regional contractile function (i.e. wall motion) and remodeling indices (i.e. localized thinning). Additionally, the authors state "at one week after ischemia ... cine MRI ... suggesting that the subacute infarct swine model did not show ... substantial ... edema". Myocardial

edema is typically assessed by T2 imaging: Was quantitative T2 data available to conform lack of edema? (if not, this statement should be revised)

- Results section indicates that among animals “iron concentrations in infarct regions were elevated compared to remote myocardium in all groups” whereas magnetic susceptibility of animals with transient 45 minute occlusion was not substantially elevated (0.01 ± 05). Data analysis could be strengthened by testing specific QSM cutoffs (via AUC curves or appropriate methods) as criteria for histology evidenced increased iron content; comparisons could be made to conventional (T2*) MRI data.
- It is difficult to understand the explanation for T2*/QSM discordance in longer term occlusion (permanent MI group) compelling. The authors state “unexpectedly, magnetic susceptibility, but not T2*, was significantly different within the infarct of the permanent MI group ... this suggests that T2* may not be sensitive to small changes in iron content, or that T2* may not decrease despite elevated iron content observed on ICP-OES”: Histopathology analysis suggests that iron content was highest in the permanent MI group ($0.12 \pm .14$), which makes pathophysiologic sense based on known concepts related to IMH following AMI: If this is the case, its not that T2* was missing small amounts of iron, but instead that it was negative in the group with highest amount of histopathology evidenced iron – this seems counterintuitive, and suggests that technical factors (or potential sources of bias) may have confounded results.
- Given that the authors postulate that “edematous inflammation and early replacement fibrosis” (page 11) may have explained discordance between T2* and histopathology evidenced iron content, the study could be strengthened by additional analysis testing whether negative T2* was more common in large animal models with histopathology evidence of such findings.
- Clinical analyses use T2* as the reference for post-AMI IMH, despite above noted large animal data suggesting that T2* may be limited for this purpose. While the authors approach is logical (given than an alternative standard for in-vivo hemorrhage imaging is not available), analysis could be strengthened by presenting text reporting whether QSM results varied in relation to surrogate markers of infarct severity that could impact risk for IMH: For example, did QSM vary in relation to duration of chest pain onset ☐ revascularization, door to balloon time, infarct transmuralty or no reflow zone on LGE?
- If feasible, the authors should consider testing MVO size (in animals and patients) in relation to QSM results.

- Clinical images were acquired at 1.5T, whereas large animal was performed at 3T. Large animal studies were performed 1 week post-MI, whereas clinical studies were performed 2.9 ± 1.5 days post MI: Differences in scanner field strength and timing of post-MI imaging (as well as in vivo vs. ex vivo data acquisition noted above) undermines optimal comparisons between animal model and human data.

- Image analysis states that full width half maximum method was used for infarct quantification: How did the authors deal with areas of MVO (which would presumably be not encompassed within areas of MI based on the FWHM technique), and what standardized criteria were used to define MVO?

- Image acquisition detail for in vivo human imaging. "T2*-

weighted multi-echo gradient-echo images were obtained at $1.8 \times 1.8 \times 0.7$ mm³ resolution". Is this 3D with navigator respiratory gating also used, or breathhold 2D that would have misregistration issue among breathholds. Also the image quality (Fig.S8) seems lack of the right-left ventricle contrast between deoxy- and oxy-heme blood pools.

- What about regression/correlation/concordance between susceptibility values and ICP measured iron values (Fig.1b and 1c)?

Minor Issues

- Results (paragraphs 1, 2) refers to imaging "one week after ischemia" and the term "ischemia" is subsequently used throughout the paper: Given that this is a post-MI model for which coronary ligation is being performed, the term "occlusion" (rather than ischemia) is more appropriate.

- Table S1: what is the row 1 name? NA entry does not make sense, as these quantitative values can be measured.

- Table S3 reports hemorrhage in a binary manner (yes/no): Was a binary criterion used to establish presence of hemorrhage on T2* weighted imaging? If so, the cutoff should be stated (and justified

by prior literature or other rationale). Additionally, a footnote could be added to the Table S3 to specify this cutoff.

Reviewer #2 (Remarks to the Author):

This manuscript describes a new non-invasive methodology - magnetic susceptibility measurements - for the quantification of iron in the pathological heart, following ischemia, without and with manipulations. Using this methodology in large animals and human patients the Authors could demonstrate a direct relationship between the levels of tissue iron after ischemia followed by short reperfusion, on one hand, and the duration of the ischemia, on the other, in the range of 45-180 min, and at 7 days (without reperfusion). In patients they showed some degree of reverse correlation between iron levels and the degree of success of treatment the patient had undergone.

This is a very important and timely subject. There is a need for new methodologies, preferably non-invasive, for monitoring the status of iron and its nature, in tissues. Otherwise, the specific data shown and the correlation between elevated tissue iron levels and tissue injury is of confirmatory value; these were already shown in the heart and brain.

Comments:

Iron is an essential element that is an important part in the activity of numerous proteins. On the other hand, iron, in particular the 'labile and redox-active fraction' of the total iron, has been incriminated as a major factor in tissue injury, even under normal iron status. The enthusiasm of this reviewer would have been markedly enhanced if the Authors would have investigated the contribution of their new technique to the understanding of the nature of the iron and its source, as they discuss (theoretically) in the Discussion Section.

In the isolated heart, in a blood-less system (and without the involvement of immune-related cells and factors) it was shown that reperfusion injury occurs in the absence of an external source of iron.

Just by re-distribution of the present tissue-iron. A strong correlation between the duration of the ischemia and the extent of iron mobilized within and out of cells, and in particular with the fraction of the labile iron was demonstrated. Also, the degree of iron re-distribution and mobilization was proposed to serve as a bona fide indicator of the degree of reperfusion injury of the ischemic and reperfused heart. In general it can be stated that often the total amount of iron in cell or tissue does not represent the extent of tissue injury it exerts. For clarity it is reiterated that the total level of iron does not govern the degree of injury it inflicts. It is somewhat disappointing that the Authors have not referred to this small, but highly reactive, fraction of labile iron. It was also demonstrated that once the heart gets a signal about 'local' excessive levels of iron (like under pre-conditioning), it forms high amount of new (apo) ferritin that is capable to scavenge any new labile iron and arrest its redox activity, thus, minimizing the production of deleterious ROS, and the consequent loss of function (or interference with salvaging processes).

Did the Authors consider that the elevation in the levels of iron in the previously ischemic (and mechanically disrupted) regions of the heart could have experienced local hemorrhage (extravagation of minute amount of the iron-rich erythrocytes)? And that this iron level can increase during the induced ischemia? But decrease during long period afterwards?

Please see:

Krause GS et al., Ann Emerg Med November 1985;14: 1037-1043.]

Kruszewski M, Labile iron pool: the main determinant of cellular response to oxidative stress. Mutat Res. 2003;531(1-2):81-92.

Chevion M et al., Copper and iron are mobilized following myocardial ischemia: Possible predictive criteria for tissue injury. Proc. Natl. Acad. Sci. USA Vol. 90, pp. 1102-1106, February 1993

Reviewer #3 (Remarks to the Author):

This manuscript describe an MRI approach for imaging of post-infarcts iron using animal model and patient. The topic is of clinical importance, however there are several issues with the study.

1. If I understand this correctly, the animal images for QSM is ex-Vivo. If this is indeed correct, this is a major limitations and the study provide no more useful information. Of ourselves iron will impart QSM signal and several prior studies have already demonstrated this in different tissue types and this study provide only incremental info.

2. It is very difficult to understand how they match tissue location in MRI to histology slies.

3. Patient study is very preliminary with small number of subjects. Image quality is very poor, one can simply identify other areas with abnormal QSM signal. There is no standard of referene in human to validate any of the claim.

4. There is lack of reproducibility for the imaging and analysis which significantly redues reliability of the data.

5. As authors pointed out QSM in heart is quite challenging and there are many confounders that impact the measurements, without a proper validation using in-Vivo imaging in animal and rigorously demonstrating a validated technique, there is very little value in this study.

NCOMMS-19-00685: Response to the Reviewers
Iron imaging in myocardial infarction reperfusion injury

RE: NCOMMS-19-00685

Dear Reviewers,

We greatly appreciate the opportunity to submit this revised manuscript to Nature Communication for review. In our response to the reviewers, we have addressed point-by-point the reviewers concerns and included changes to the manuscript. We wish to highlight several additional experiments we performed that address the major concerns of the reviewers that required additional time and caused a delay in resubmission:

- R1. We investigated temporal changes in QSM by performing a cross-sectional study at 1 (n=15) and 8 weeks (n=8) after infarction. This data shows a reduction magnetic susceptibility between 1 and 8 weeks consistent with iron, histology and assays.
- R1. We show that findings in humans are replicated in a large animal study in the first 3 days post-infarction.
- R2. We quantified contributions from protein-bound or labile iron in the infarct region by performing additional assays targeting labile (catalytic) iron using electron paramagnetic resonance (EPR), and protein-bound iron using messenger RNA analysis of ferritin response. This data shows that there was an elevation in labile iron in the infarct region.

Again, we are grateful to the reviewers for their very constructive and insightful comments that we believe have led to an improved manuscript.

Many thanks,

Walter Witschey, Ph.D.
Assistant Professor of Radiology
Perelman School of Medicine
University of Pennsylvania

Comments from Reviewer #1

- 1. A key limitation concerns the fact that QSM in the large animal model was performed post-mortem (*ex vivo*), meaning that sacrifice itself could have induced changes in myocardial tissue properties (e.g. infarct expansion). This differs from the human (clinical) protocol, and should be addressed as a source of bias and potential challenge with respect to co-localization.**

R1 points to the limitations of extrapolating *ex vivo* swine data to the *in vivo* human condition. To address these concerns, we undertook the following additional experiments:

- We performed additional *in vivo* QSM experiments within 2 days of occlusion in 3 pigs to align better with the human experiments. The data is consistent with the human condition as shown in supplementary figures: Fig. S4 and S9.
- We show *in vivo* QSM data from several animals and that the *in vivo* condition also has alterations in magnetic susceptibility (Fig. 5 and S9). This is consistent with our findings in humans (Fig 6 and S10).
- Although we found close agreement between animals and humans at the same time point, nevertheless sacrifice will potentially result in changes between *in vivo* and *ex vivo* conditions, including potential changes in infarct size due to loading conditions, loss of contractile function, and rigor mortis.
 - To reduce some of these anticipated changes as much as possible, we sought to perform *ex vivo* MRI within 1-2 hours following sacrifice of the animal.
- We included supplementary data showing good correspondence between *in vivo* and *ex vivo* QSM (Fig. S9).

- 2. It should also be noted in large animal model, cardiac QSM was performed 1 week after occlusion: Rationale for this time point is unclear.**

To address this concern, we performed additional experiments in 3 animals at the same time post-infarction as the humans studies (Fig. S9). This new data shows that the animal and human findings are consistent.

A combination of factors led to the initial decision to perform MRI at 1-week post-infarction:

- We have previous experience with the 1-week time point for MRI and tissue analysis (Stoffers 2017, Witschey 2012) and sought to build upon this knowledge.
- The animal studies were initiated prior to designing the human clinical study. When the results of the animal studies showed promise, we sought to collect human MRI data. In early conversations with our co-author Dr. Matthai at Penn Presbyterian Hospital, it was advised that almost all patients would be discharged by 3 days after their reperfusion procedure and that it would be much more difficult to recruit patients for an out-patient MRI at 1 week. The design of the human study was to recruit STEMI patients for an in-patient MRI at a high rate.

- 3. Lack of serial (*in vivo*) imaging in large animal model limits insight into whether IMH would have been larger had animals been imaged earlier after AMI, or whether temporal time course of QSM change parallels LV remodeling patterns (i.e. LV dilation and/or wall thinning).**

As discussed, we undertook an additional study to investigate the progression of iron and magnetic susceptibility post-infarction. We found a reduction in iron and magnetic susceptibility between 1 and 8 weeks. Fig 5 shows representative 1 and 8 week MRI images from one animal along with quantitative results from all. Fig. S9 shows imaging results from all animals. The new section is labeled “*Magnetic susceptibility and iron evolve during wound healing*”

4. Animal model presents global indices of LV function (e.g. EF, volumes): Data could be further strengthened by including regional contractile function (i.e. wall motion) and remodeling indices (i.e. localized thinning).

We have included regional analysis including wall thickening, ED and ES wall thickness and wall motion (Medis QMass). As might be expected, these results show both temporal and spatial differences between different time points (baseline vs. 1 or 8 weeks) post-infarction and regions (infarct vs. remote) (see Table S2).

5. Additionally, the authors state “at one week after ischemia ... cine MRI ... suggesting that the subacute infarct swine model did not show ... substantial ... edema”. Myocardial edema is typically assessed by T2 imaging: Was quantitative T2 data available to conform lack of edema? (if not, this statement should be revised)

We did not quantify edema using T2 imaging, so we removed this statement.

6. Results section indicates that among animals “iron concentrations in infarct regions were elevated compared to remote myocardium in all groups” whereas magnetic susceptibility of animals with transient 45 minute occlusion was not substantially elevated (0.01 ± 05). Data analysis could be strengthened by testing specific QSM cutoffs (via AUC curves or appropriate methods) as criteria for histology evidenced increased iron content; comparisons could be made to conventional (T2*) MRI data.

We performed receiver operating characteristic (ROC) analysis to test the sensitivity and specificity of QSM to iron and compare it to conventional MRI (T2* and R2*). This analysis shows that magnetic susceptibility is comparable to T2* and R2* in detection of iron. However, we believe this analysis is problematic because there is overlap in the probability distributions for iron between the 45 min groups and the remote myocardium due to experimental error in histologically separating infarct and remote myocardium in non-transmural infarctions in the 45 minute group. In the ROC analysis, the histologic gold standard iron measurement is transformed from a continuous variable into a binary variable using different cutoffs. Consequently, a cutoff taken as 1 or 2 standard deviation elevation from remote myocardial iron concentration causes the binary classifier to mislabel infarcted regions as non-infarcted. These regions, which show elevated magnetic susceptibility, become false positives in the ROC analysis when using higher iron cutoffs. We believe that further refinement in histological sampling for ICP is required to set the gold standard for comparison to MRI.

We alternately investigated receiver operating characteristic (ROC) using the infarct region from T2*w MRI based on thresholding hyperintense and hypointense (when present) regions. Tissue magnetic susceptibility provided AUC = 0.92 and at the optimal operating point sensitivity = 0.91 and specificity = 0.86. T2* and R2* ROC analysis provided an AUC= 0.71 and at the optimal operating point sensitivity = 0.65 and specificity = 0.90. Tissue magnetic susceptibility showed superior balance between sensitivity and specificity at classifying infarct versus remote viable myocardium compared to T2* and R2* (Fig. S10). To be clear, this analysis is not based on MRI and not histological findings so does not have a genuine orthogonal gold standard. However, it is in good agreement with the association testing between iron and MRI, which shows that, there is good correlation between QSM/T2*/R2* in 90 and 180 min animal groups, but that T2*/R2* correlations fall when including other groups (Fig. S11).

- It is difficult to understand the explanation for T2*/QSM discordance in longer term occlusion (permanent MI group) compelling. The authors state “unexpectedly, magnetic susceptibility, but not T2*, was significantly different within the infarct of the permanent MI group ... this suggests that T2* may not be sensitive to small changes in iron content, or that T2* may not decrease despite elevated iron content observed on ICP-OES”: Histopathology analysis suggests that iron content was highest in the permanent MI group ($0.12 \pm .14$), which makes pathophysiologic sense based on known concepts related to IMH following AMI: If this is the case, its not that T2* was missing small amounts of iron, but instead that it was negative in the group with highest amount of histopathology evidenced iron – this seems counterintuitive, and suggests that technical factors (or potential sources of bias) may have confounded results.

The reason for the discrepancy between $T2^*$ compared to total iron/magnetic susceptibility measurements is not yet fully clear and may be caused by technical factors or an incomplete understanding of the physical mechanisms underlying $T2^*$ contrast in their relation to infarct pathophysiology. There was only a moderate association between $R2^*$ and iron ($R^2 = 0.19$, $P < 0.05$), but this improved when excluding the 45-min and permanent infarct group ($R^2 = 0.58$, $P < 0.001$). It has been observed that there is an association between the duration of ischemia, mobilization of iron in and out of cells and the labile iron fraction, and time since the initial event. A differential relaxation rate enhancement between reperfused and permanent infarctions could occur if, in the reperfused group, there were an increased rate of exchange between water and water in close association with proteins that sequester iron in cells or labile iron or a spatially localized region of these proteins or labile iron concentrated in tissues. It was recently shown in patients that the amount of hemorrhage peaks on day 2 post-MI using $T2^*$ MRI. Technical factors such as image spatial resolution affect $T2^*$ relaxation times. In the presence of local field gradients, a large voxel has a shortened $T2^*$ relaxation time compared to a small voxel. The physical principle here is that $T2^*$ is predominantly caused by intravoxel dephasing associated with local magnetic field gradients in the voxel. A larger voxel is subject to a wider distribution of local magnetic fields – these will tend to cause the net MR signal phase in the voxel to dephase more rapidly.

To better understand the discrepancy between $R2^*$ and iron, we obtained high resolution images of the infarct region using a 7 T MRI scanner and compared them to histology. *Please note that we would prefer this data to remain unpublished in anticipation of a follow-up submission to the journal Magnetic Resonance in Medicine.* (A) shows the MR images acquired at 3 T (0.9 mm isotropic resolution) and at 7 T (0.5 mm isotropic resolution). The 180-min reperfused infarct shows distinct $T2^*$ -w and $T2^*$ -map hypointense regions which correspond to an elevation in $R2^*$. QSM show regions of a paramagnetic shift in both the reperfused and permanent (non-reperfused) infarcts. (B) permanent infarct magnetic susceptibility at high resolution 7 T localizes to the iron distribution seen with histology. There is a lack of hypointense regions seen within the infarct at 3 T. At 7 T, there is a change in contrast at the epicardial border of the infarct (red arrow) which correspond to iron distribution at the transition zone between necrotic and mixed viable myocytes (Prussian blue) and trapped red blood cells (H&E). Histology shows extensive fibrosis (Trichrome, yellow arrows) which correspond to regions containing iron, RBCs and elevated magnetic susceptibility. The level of detail in spatial distribution of tissue magnetic susceptibility appears to be compromised when increasing voxel size at 3 T.

8. Given that the authors postulate that “edematous inflammation and early replacement fibrosis” (page 11) may have explained discordance between T2* and histopathology evidenced iron content, the study could be strengthened by additional analysis testing whether negative T2* was more common in large animal models with histopathology evidence of such findings.

Is the absence of a hypointense region associated with early replacement fibrosis instead of hemorrhage? We believe that the data shown in Fig. 3C together with the high resolution images (7 T above) show that they do associate (this is a spatial correspondence), although as indicated by the magnetic susceptibility and iron assays, absence of hypointense region does not rule out the presence of iron. In fact, some regions that do not show much hypointensity at 3 T seem to show more at 7 T (higher R2*, see white arrow above), suggesting the higher field induces a much stronger field in these regions. High resolution images show that infarcts are highly spatially heterogeneous at the sub-acute time point in both the reperfused and permanent infarctions.

Replacement fibrosis has been shown to increase transverse relaxation times throughout permanently infarcted myocardium in the subacute phase (see e.g. Stoffers 2017 J. Cardiovasc. Magn. Reson).

We did not look at the association with edema so we removed the in-line reference to it.

- 9. Clinical analyses use T2* as the reference for post-AMI IMH, despite above noted large animal data suggesting that T2* may be limited for this purpose. While the authors approach is logical (given than an alternative standard for in-vivo hemorrhage imaging is not available), analysis could be strengthened by presenting text reporting whether QSM results varied in relation to surrogate markers of infarct severity that could impact risk for IMH: For example, did QSM vary in relation to duration of chest pain onset revascularization, door to balloon time, infarct transmuralty or no reflow zone on LGE?**

The duration from symptom onset to reperfusion was 303.7 min in the no hemorrhage group and 695.5 min in the hemorrhage group (Table S3). 3 of 4 patients who showed a magnetic susceptibility alteration had transmuralty>75% and MVO> 5% LV mass. Of the patients who we did not detect a change in magnetic susceptibility, none had transmuralty >75% and only 1 had MVO>5%.

- 10. If feasible, the authors should consider testing MVO size (in animals and patients) in relation to QSM results.**

We found a significant positive association between MVO size and magnetic susceptibility in animals. These results are provided in supplementary data Fig. S10. In the permanent group, we were not able to detect a hypointense region on T2* despite significant MVO and changes in magnetic susceptibility. This suggests that while there are associations between T2*, LGE and magnetic susceptibility, they are sensitive to different pathology. We are not sufficiently powered to detect this association in humans although we found 3 of 4 patients who showed a magnetic susceptibility alteration had MVO> 5% LV mass.

- 11. Clinical images were acquired at 1.5T, whereas large animal was performed at 3T. Large animal studies were performed 1 week post-MI, whereas clinical studies were performed 2.9 ± 1.5 days post MI: Differences in scanner field strength and timing of post-MI imaging (as well as in vivo vs. ex vivo data acquisition noted above) undermines optimal comparisons between animal model and human data.**

To address the concern of timing post-MI, we conducted a study in 3 animals at the 3 day time point and found that the new animal findings also show alterations in magnetic susceptibility similarly to humans (see Fig. S4 and Fig. S9).

While the magnetic susceptibility is independent of field strength, the field does affects the way it is measured. A larger applied magnetic field induces a larger local magnetic field and affects the MR signal phase. Generally, a low field (1.5 T) measurement requires a larger echo time to achieve the same phase sensitivity as a high field measurement. We acknowledge this as a limitation of the animal and human comparisons.

- 12. Image analysis states that full width half maximum method was used for infarct quantification: How did the authors deal with areas of MVO (which would presumably be not encompassed within areas of MI based on the FWHM technique), and what standardized criteria were used to define MVO?**

After defining the infarct region with FWHM, microvascular obstruction (MVO) regions were defined as a hypointense core within a hyperenhancing region on LGE and were manually

segmented in Medis QMass. This approach is consistent with the literature (see e.g. Mangion, et al. Eur. J. Radiol. 2019).

13. Image acquisition detail for in vivo human imaging. "T2*-weighted multi-echo gradient-echo images were obtained at 1.8x1.8x0.7 mm³ resolution". Is this 3D with navigator respiratory gating also used, or breathhold 2D that would have misregistration issue among breathholds. Also the image quality (Fig.S8) seems lack of the right-left ventricle contrast between deoxy- and oxy-heme blood pools.

Multiecho gradient-echo images were obtained by 2D breathhold acquisition. This can cause misalignment between images at different slice positions, especially for patients who are not consistent about their respiratory position. We sought to reduce misalignment by training the patient to perform consistent breathholds in advance of the scan.

Wen Y (2017 Magn. Reson. Med) reported a change in RV-LV contrast using QSM due to the difference in magnetic susceptibility of venous and arterial blood. We did not observe this finding in all patients or pigs, although we suspect this may in part be due field heterogeneity or misalignment between slice positions. Notably Wen, et al. also found that 36% of patients had uninterpretable QSM image quality. It is not entirely clear from their description what this means, but we think it also implies that the RV susceptibility was not always observed for reasons of thick slices, long scan time, breath-hold, and cardiac phase inconsistency.

The 1.5 T field strength could be the cause of a lack of right-left ventricle contrast between deoxy- and oxy-heme blood pools, the low field strength may not cause enough intravoxel dephasing in the RV (deoxy-Hb) blood pool to detect an elevation in susceptibility seen with QSM. Voxel size also affects the magnetic susceptibility and could be causing a decrease in susceptibility within the RV caused by increased averaging of individual proton spins with increased voxel size. For *in vivo* animal studies at 3 T and higher resolution, we are seeing a contrast in the right-left ventricle due to deoxy- and oxy- blood pools (Fig. 5 and S9).

We anticipate that it will be possible to develop solutions to many of these issues using faster scanning (acceleration or multiband excitation), 3d acquisitions, and navigation.

14. What about regression/correlation/concordance between susceptibility values and ICP measured iron values (Fig.1b and 1c)?

We conducted simple linear regression of magnetic susceptibility and total iron correlation concentration and found significant correlation between the two. Details as shown in Supplementary Fig. S10B.

15. Results (paragraphs 1, 2) refers to imaging “one week after ischemia” and the term “ischemia” is subsequently used throughout the paper: Given that this is a post-MI model for which coronary ligation is being performed, the term “occlusion” (rather than ischemia) is more appropriate.

We changed the term “ischemia” to “occlusion” throughout the manuscript.

16. Table S1: what is the row 1 name? NA entry does not make sense, as these quantitative values can be measured.

Row 1 was adjusted to be “time to reperfusion”. NA was added to cells that did not have a measurement. For example, there were no hypointense regions seen in the 45-min and permanent ligation infarcts, therefore, we could not obtain a susceptibility or T2* measurement with Hypo regions.

17. Table S3 reports hemorrhage in a binary manner (yes/no): Was a binary criterion used to establish presence of hemorrhage on T2* weighted imaging? If so, the cutoff should be stated (and justified by prior literature or other rationale). Additionally, a footnote could be added to the Table S3 to specify this cutoff.

Hemorrhage (yes/no) in new Table S4 was defined as a hypointense area >1 g tissue in T2*-weighted image corresponding to the region of myocardial infarction as assessed by late gadolinium enhanced MRI. This is consistent with other literature definitions such as in Carrick, et al. 2015.

Comments from Reviewer #2

Iron is an essential element that is an important part in the activity of numerous proteins. On the other hand, iron, in particular the 'labile and redox-active fraction' of the total iron, has been incriminated as a major factor in tissue injury, even under normal iron status. The enthusiasm of this reviewer would have been markedly enhanced if the Authors would have investigated the contribution of their new technique to the understanding of the nature of the iron and its source, as they discuss (theoretically) in the Discussion Section.

In the isolated heart, in a blood-less system (and without the involvement of immune-related cells and factors) it was shown that reperfusion injury occurs in the absence of an external source of iron. Just by re-distribution of the present tissue-iron. A strong correlation between the duration of the ischemia and the extent of iron mobilized within and out of cells, and in particular with the fraction of the labile iron was demonstrated. Also, the degree of iron re-distribution and mobilization was proposed to serve as a bona fide indicator of the degree of reperfusion injury of the ischemic and reperfused heart. In general it can be stated that often the total amount of iron in cell or tissue does not represent the extent of tissue injury it exerts. For clarity it is reiterated that the total level of iron does not govern the degree of injury it inflicts. It is somewhat disappointing that the Authors have not referred to this small, but highly reactive, fraction of labile iron.

It was also demonstrated that once the heart gets a signal about 'local' excessive levels of iron (like under pre-conditioning), it forms high amount of new (apo) ferritin that is capable to scavenge any new labile iron and arrest its redox activity, thus, minimizing the production of deleterious ROS, and the consequent loss of function (or interference with salvaging processes).

Did the Authors consider that the elevation in the levels of iron in the previously ischemic (and mechanically disrupted) regions of the heart could have experienced local hemorrhage (extravagation of minute amount of the iron-rich erythrocytes)? And that this iron level can increase during the induced ischemia? But decrease during long period afterwards?

We would like to thank the reviewer for the suggestion to investigate the source of iron in the infarct area. We undertook a series of analyses aimed to verify the contribution of both intracellular labile/catalytic iron and test whether hemorrhage contributes to increased iron. We believe these additional assays strengthen our manuscript greatly.

We have added the following test to the results.

We further investigated the nature of tissue iron, including labile (redox-active) iron concentration using electron paramagnetic resonance (EPR), gene expression assays and immunohistochemistry. EPR measurements showed significantly increased labile iron in infarct region (Fig. 2 A and S8 B). To confirm the increase of labile iron, we tested the gene expression of cellular iron metabolism markers in infarct (n=24) and remote myocardium (n=20) tissue samples. Ferritin is a scavenger of intracellular labile iron, arresting its redox activity. Ferritin light chain (FLC) expression significantly increased in the infarct (infarct, 1137.2% Myo vs. remote myocardium, 88.1% Myo) (Fig. 2 B). Ferritin heavy chain (FTH1) expression was not

significantly modified in infarct vs. remote area in any of the conditions studied (Fig. 2 B). The expression of the intracellular iron sensor Iron Regulatory Protein 2 (IRP2, also known as ACO1) was significantly decreased in the infarct region (infarct, 61.4% Myo vs. remote myocardium, 102.4% Myo) (Fig. 2 C). Divalent metal transporter 1 (DMT1), which transports iron into the labile iron pool, was significantly decreased in the infarct region (infarct, 53.7% Myo vs. remote myocardium, 99.1% Myo) (Fig. 2 D). Heme oxygenase-1 (HO1) transcription is activated in response to increased intake of heme proteins, and its expression was significantly increased (infarct, 1350.3% Myo vs. remote myocardium, 100.5% Myo) (Fig. 2 E). There was increased hemoglobin staining by immunohistochemistry in reperfused and permanently occluded animals, compared to remote myocardium (Fig. 2 F,G).

Comments from Reviewer #3

- 1. If I understand this correctly, the animal images for QSM is ex-Vivo. If this is indeed correct, this is a major limitations and the study provide no more useful information. Of ourselves iron will impart QSM signal and several prior studies have already demonstrated this in different tissue types and this study provide only incremental info.**

We agree that extrapolation of ex vivo pig results to the human condition has inherent limitations. To address this concern, we implemented *in vivo* QSM in the pig cohort at 1-week (N=6) and 8-weeks (N=6) post-MI. The in vivo swine results (Fig 5 and S9) show elevation in susceptibility within the infarct region. The location of the infarct was demonstrated with hyperintense regions seen with late gadolinium enhanced imaging. In consideration of comments from Reviewer 1, we have also studied an additional 3 animals at the same subacute time point (3 days) as the human study and find that these results appear to be replicated.

As of now, there is only one other publication on cardiac QSM (Wen, et al. 2017 Magn. Reson. Med.) and the study reported magnetic susceptibility of venous blood in the right ventricle. Our study is the first to detect intramyocardial hemorrhage in both an animal and human studies using QSM. Additionally, we show several novel findings about the underlying pathology including the mismatch between T2* and magnetic susceptibility in the infarction. Further investigation showed that there is significant iron contributing to the magnetic susceptibility map in the infarct region, even when these changes are not appearing on conventional MRI.

- 2. It is very difficult to understand how they match tissue location in MRI to histology slices.**

Tissue fiducial markers such as the RV insertion point, papillary muscles, and long axis dimensions of the heart were used to correlate MR images with histology. A critical preliminary step in this analysis is the sectioning of the heart into short axis section parallel to the imaging planes. It is not a direct match between MRI and histology and we do not do imaging of the histological sections. Data such as is shown in Fig 1D, E shows associations between imaging contrast and histological evidence of fibrosis and iron deposition.

- 3. Patient study is very preliminary with small number of subjects. Image quality is very poor, one can simply identify other areas with abnormal QSM signal. There is no standard of referene in human to validate any of the claim.**

The limitations of human in vivo cardiac QSM and improvements to GRE acquisition are explained in R1 response 10. To validate the elevation in susceptibility seen in vivo is indeed due to an elevation in iron content, we obtained in vivo QSM in an animal cohort (R3, comment 1) where we saw an elevation in susceptibility detected in vivo was associated with an increase in iron concentration (Fig. 5)

- 4. There is lack of reproducibility for the imaging and analysis which significantly reduces reliability of the data.**

Reproducibility of analysis was tested between two raters (Fig. S12). Although we did not test imaging reproducibility due to scan time restrictions on patients and animals, we validated the elevation in susceptibility seen in vivo animal studies with ex vivo QSM and iron content analysis.

- 5. As authors pointed out QSM in heart is quite challenging and there are many confounders that impact the measurements, without a proper validation using in-Vivo imaging in animal and rigorously demonstrating a validated technique, there is very little value in this study.**

We successfully obtained in vivo QSM in a cohort of animals to validate the findings seen in STEMI patients, please see additional figures (Fig 5, S4, S9).

Reviewers' comments:

Reviewer #2 (Remarks to the Author):

The Authors have conducted additional experiments, of value, which improve the overall quality of their manuscript. However, there are more than few misinterpretations and incorrect descriptions/citations, which do not allow for its acceptance for publication.

COVER LETTER – Response to R2:

1. "R2. We quantified contributions from protein-bound or labile iron in the infarct region performing additional assays targeting labile (catalytic) iron using EPR and protein-bound iron using mRNA analysis of ferritin response. These data show that there was an elevation of iron in the infarct region."

EPR measures mon-nuclear ferric iron, whether it is protein-bound or in complex with small molecular weight molecules. This statement is confusing and incorrect.

2. "We would like to thank the reviewer for the suggestion to investigate the source of iron in the infarct area. We undertook a series of analyses aimed to verify the contribution of both intracellular labile/catalytic iron and test whether hemorrhage contributes to increased iron. We believe these additional assays strengthen our manuscript greatly. We have added the following test to the results. We further investigated the nature of tissue iron, including labile (redox-active) iron concentration using electron paramagnetic resonance (EPR) [R2 – see response to Comment #1, above] -, gene expression assays and immunohistochemistry. EPR measurements showed significantly increased labile iron in infarct region (Fig. 2 A and S8 B). To confirm the increase of labile iron, we tested the gene expression of cellular iron metabolism markers in infarct (n=24) and remote myocardium (n=20) tissue samples. Ferritin is a scavenger of intracellular labile iron, arresting its redox activity. Ferritin light chain (FLC) expression significantly increased in the infarct (infarct, 1137.2% Myo vs. remote myocardium, 88.1% Myo) (Fig. 2 B). Ferritin heavy chain (FTH1) expression was not significantly modified in infarct vs. remote area in any of the conditions studied (Fig. 2 B). The expression of the intracellular iron sensor Iron Regulatory Protein 2 (IRP2, also known as ACO1) was significantly decreased in the infarct region (infarct, 61.4% Myo vs. remote myocardium, 102.4% Myo) (Fig. 2 C). Divalent metal transporter 1 (DMT1), which transports iron into the labile iron pool, was significantly decreased in the infarct region (infarct, 53.7% Myo vs. remote myocardium, 99.1% Myo) (Fig. 2 D). Heme oxygenase-1 (HO1) transcription is activated in response to increased intake of heme proteins, and its expression was significantly increased (infarct, 1350.3% Myo vs. remote myocardium, 100.5% Myo) (Fig. 2 E). There was increased hemoglobin staining by immunohistochemistry in reperfused and permanently occluded animals, compared to remote myocardium (Fig. 2 F,G)."

See answer to Comment \$1, above.

BODY OF THE REVISED MANUSCRIPT:

3. Abstract – “Magnetic susceptibility changes were associated with elevated total and catalytic labile iron using histology, inductively coupled plasma optical emission spectrometry and electron paramagnetic resonance (EPR) spectroscopy.”

None of the techniques/methodologies mentioned in this sentence can be used for the quantitation of labile iron, including EPR.

4. Page 7 - “We further investigated the labile(redox-active) iron concentration using electron paramagnetic resonance (EPR), ...”

EPR can be used to quantitate mono-nuclear (versus multimeric or aggregated) ferric iron, including in small mono-iron ferric complexes and proteins (like met-hemoglobin but not normal ferro-Hb). These EPR detectable species contain not only ferric redox active iron.

5. Page 7 – “To confirm the increase of labile iron, we tested the gene expression of cellular iron metabolism markers in infarct (n=24) and remote myocardium (n=20) tissue samples. Ferritin is a scavenger of intracellular labile iron, arresting its redox activity. Ferritin light chain (FLC) expression significantly increased in the infarct (infarct, 1137.2% Myo vs. remote myocardium, 88.1% Myo) and ferritin heavy chain (FTH1) expression was not significantly modified in infarct vs. remote area in any of the conditions studied (Fig. 2 B). “ [Underline R2]

The underlined part could be the most interesting finding in this paper. Ferritin composition varies among the organs, in the same organism, reflecting its organ-specific function. In the heart, the ferritin heavy chain is the dominant subunit (~70%). The Authors found that in the infarcted area the message for the ferritin light chain is the dominant component. The change in the composition of ferritin must reflect an important change in the function of this ‘new’ ferritin.

6. Page 7 – “The expression of the intracellular iron sensor Iron Regulatory Protein 2 (IRP2, also known as ACO1) was significantly decreased in the infarct region (infarct, 61.4% Myo vs. remote myocardium, 102.4% Myo) (Fig. 2 C). Divalent metal transporter 1 (DMT1), which transports 8 iron into the labile iron pool, was significantly decreased in the infarct region (infarct, 53.7% Myo vs. remote myocardium, 99.1% Myo) (Fig. 2 D). Heme oxygenase-1 (HO1) transcription is activated in response to increased intake of heme proteins, and its expression was significantly increased (infarct, 1350.3% Myo vs. remote myocardium, 100.5% Myo) (Fig. 2 E). There was increased hemoglobin staining by immunohistochemistry in reperfused and permanently occluded animals, compared to remote myocardium (Fig. 2 F,G and S7), which contributed to the increased HO1 expression.”

IRP 1 binds to the stem-loop structure of the message of iron proteins; in the presence of labile iron IRP 1 changes its tertiary structure (upon binding of ferric iron) and turns into aconitase 1. NOT IRP 2.

The increase of HO-1 could be associated with the increase in the level of denatured hemoglobin (metHb?), and the need to degrade the heme 'ring' and release its iron.

The numbers depicted along the entire manuscript do not represent the 'real' degree of significance of the measurements; i.e., 1350.3% increase. Do the Authors mean that their accuracy is valid across 5 digits?

7. Page 12 – “Elevated magnetic susceptibility shown on QSM was associated with accumulation of post-infarction tissue iron, increased labile iron ...”

This association was not established.

8. Page 14 – ‘iron responsible elements’- should be iron responsive elements.

9. Page 24 – “Electron Paramagnetic Resonance spectroscopy (EPR)”.

The procedure described is intended to convert all the iron in the sample into ferric (3+) iron, through oxidation of all iron by desferrioxamine B. Only comparison with O-phenanthroline could allow for the quantitation of the ferric and ferrous iron.

Reviewer #4 (Remarks to the Author):

The authors have addressed the concerns raised by the previous reviewers in detail with additional experiments and clarifications. This is one of the few works where QSM measurements are independently validated and the authors are to be recommended for their study. I have only one minor comment.

1. (Response to Reviewer #1, Comment 13) The method used by the authors to acquire QSM in patients is indeed susceptible to misregistration between slices each acquired in a single breath-hold as the reviewer suggests. This is also the cited reason in Wen et al. (cited in the response and the

manuscript) why 36% of their patients had uninterpretable QSM quality. The authors may want to add this as a potential limitation. The same group has recently presented follow-up work using a free-breathing acquisition.

NCOMMS-19-00685: Response to the Reviewers
Iron imaging in myocardial infarction reperfusion injury

RE: NCOMMS-19-00685

Dear Reviewers,

We greatly appreciate the opportunity to submit this revised manuscript to Nature Communication for review. In our response to the reviewers, we have addressed point-by-point the reviewers concerns and included changes to the manuscript. The key points to summarize are:

- In response to reviewer 2, we corrected and clarified many of the details regarding measurement of iron.

Again, we are grateful to the reviewers for their very constructive and insightful comments that we believe have led to an improved manuscript.

Many thanks,

Walter Witschey, Ph.D.
Assistant Professor of Radiology
Perelman School of Medicine
University of Pennsylvania

Response to Reviewers' comments:

Reviewer #2 (Remarks to the Author):

COVER LETTER

1. **“Response to R2: We quantified contributions from protein-bound or labile iron in the infarct region performing additional assays targeting labile (catalytic) iron using EPR and protein-bound iron using mRNA analysis of ferritin response. These data show that there was an elevation of iron in the infarct region.” EPR measures mon-nuclear ferric iron, whether it is protein-bound or in complex with small molecular weight molecules. This statement is confusing and incorrect.**
2. **“We would like to thank the reviewer for the suggestion to investigate the source of iron in the infarct area. We undertook a series of analyses aimed to verify the contribution of both intracellular labile/catalytic iron and test whether hemorrhage contributes to increased iron. We believe these additional assays strengthen our manuscript greatly. We have added the following test to the results. We further investigated the nature of tissue iron, including labile (redox-active) iron concentration using electron paramagnetic resonance (EPR) [R2 – see response to Comment #1, above] [...]**

Response to comments 1 and 2 (COVER LETTER):

Our intention to be succinct in summarizing the new experiments in our previous cover letter resulted in a confusing and inaccurate statement. To address these concerns, we made additional changes to clarify and correct misleading or inaccurate statements regarding iron quantitation. To briefly summarize, in their initial review, R2 suggested that we should further investigate the nature of iron and its source, with a focus on the “labile and redox-active fraction”, as well as iron from hemorrhage.

We reviewed the literature for a protocol to measure labile iron and found a 2014 paper on this topic by Dr. Cabantchik (doi: 10.3389/fphar.2014.00045). In this review, it was postulated that labile cell iron is a dynamic parameter that is relevant only for living cells and that prevails under defined, spatial, temporal and environmental conditions. To reliably measure labile iron in the tissue from our study, isolation of primary cells would had needed to be performed, and this was beyond the scope of the current study.

Given this limitation, we decided to use EPR on tissue isolates treated with desferrioxamine as the best available alternative. We agree with R2 that EPR does not directly quantitate labile iron, however, in conjunction with the analysis performed on expression of cellular markers of iron and labile iron metabolism, we believe that the experiments performed indirectly reflect the status of labile iron following ischemia reperfusion injury. We believe we provide a broad picture of the nature of the tissue iron detected by QSM imaging of magnetic susceptibility through a combination of EPR/desferrioxamine experiment, mRNA analysis, hemoglobin immunohistochemistry, colorimetric analysis of Fe³⁺ and Fe²⁺, in addition to the ICP-OES results and Prussian blue staining. **In our revision, we clarify that what was measured was desferrioxamine-chelatable iron and that this is only an indirect measure of labile iron.**

As the main aim of this study was to test the applicability of QSM imaging to map reperfusion injury, we believe the tissue analysis techniques performed clearly show that QSM reflects increased tissue iron. Undoubtedly, the results we present regarding desferrioxamine-chelatable iron and changes in expression of iron metabolism related genes raise new questions about

cardiac, myocardial region-specific and cell-specific responses to iron in ischemia reperfusion injury, and we will continue investigating these topics in our next research studies.

BODY OF THE REVISED MANUSCRIPT:

3. Abstract – “Magnetic susceptibility changes were associated with elevated total and catalytic labile iron using histology, inductively coupled plasma optical emission spectrometry and electron paramagnetic resonance (EPR) spectroscopy.” None of the techniques/methodologies mentioned in this sentence can be used for the quantitation of labile iron, including EPR.

Response:

As discussed above (response to comments 1 and 2), we agree with Reviewer 2 that EPR does not directly quantitate labile iron. We have amended the abstract to remove the reference to labile iron. The text now reads as follows:

“Magnetic susceptibility changes were associated with elevated iron using histology, inductively coupled plasma optical emission spectrometry and electron paramagnetic resonance (EPR) spectroscopy.”

4. Page 7 - “We further investigated the labile(redox-active) iron concentration using electron paramagnetic resonance (EPR), ...” EPR can be used to quantitate mononuclear (versus multimeric or aggregated) ferric iron, including in small mono-iron ferric complexes and proteins (like met-hemoglobin but not normal ferro-Hb). These EPR detectable species contain not only ferric redox active iron.

Response:

As discussed above (response to comments 1 and 2), EPR does not directly quantitate labile iron and affirming so without context (i.e. in the abstract) can be misleading. However, the specific technique we performed offers, in our opinion and based on previous literature, a close indirect approximation to measuring labile or redox-active iron as reflected by desferrioxamine-chelatable iron.

As Reviewer 2 explains, many iron species are paramagnetic and EPR-detectable. Signals at $g \sim 4.3$, like the one we used, correspond to mononuclear high-spin ferric ions in sites of low symmetry which can be found in several chelates and mono-iron metalloproteins (non-transport or storage proteins), and not only ferric redox active iron. As detailed in our methods, the specific experiment we performed used the siderophore compound desferrioxamine on extracts of porcine hearts. Desferrioxamine chelates free and loosely bound iron, and previous literature indicates that Fe(III)-desferrioxamine complexes signal at $g \sim 4.3$ using 9.47 GHz EPR (doi: 10.1016/j.redox.2013.11.005, doi: 10.1074/jbc.270.2.700). The iron is referred to as “chelatable/labile iron pool” or “free, redox active iron” interchangeably.

In the 2014 study by Moser et al, the authors specify that although the properties of having unrestricted redox activity and being chelatable are not necessarily mutually inclusive, they assume that the magnitude of the labile iron pool is proportional to that of the chelatable iron pool. Transferrin also signals at $g \sim 4.3$, (doi: 10.1016/0076-6879(93)27014-8), however, it shows a different shape from the one we report (see Figure S8). When we tested the cardiac isolates in the absence of desferrioxamine, we did not detect any signal at $g \sim 4.3$. Within the spectral range we used in our experiment (1350-1850 gauss) we did not detect other EPR signals in the tissue extracts, either treated or untreated with desferrioxamine. Other blood or tissue iron species, such as Fe(III)-heme iron in met-hemoglobin ($g = 6$, doi: 10.1016/0076-6879(93)27014-8), signal outside the range we used.

In summary, we concluded that the increased iron in the infarct area (Figure 2A) we detected by EPR corresponded to desferrioxamine-bound iron that, due to being free or loosely bound within the tissue, was able to bind desferrioxamine. Desferrioxamine oxidizes Fe(II) to Fe(III), hence the desferrioxamine-bound iron detected by EPR corresponds to both original Fe(III) in the homogenate plus Fe(II) converted into Fe(III). Desferrioxamine-chelatable iron corresponds to iron that has the potential of producing the damaging hydroxyl radicals via Haber-Weiss and Fenton reactions.

Reviewer 2 is correct that generally referring to the iron measured by EPR as labile iron, outside the context of the details of the technique, can be misleading. We have amended the text for accuracy, referring to the iron fraction measured by EPR as “desferrioxamine-bound iron” or “desferrioxamine-chelatable iron” when appropriate.

5. Page 7 – “To confirm the increase of labile iron, we tested the gene expression of cellular iron metabolism markers in infarct (n=24) and remote myocardium (n=20) tissue samples. Ferritin is a scavenger of intracellular labile iron, arresting its redox activity. Ferritin light chain (FLC) expression significantly increased in the infarct (infarct, 1137.2% Myo vs. remote myocardium, 88.1% Myo) and ferritin heavy chain (FTH1) expression was not significantly modified in infarct vs. remote area in any of the conditions studied (Fig. 2 B). “ The underlined part could be the most interesting finding in this paper. Ferritin composition varies among the organs, in the same organism, reflecting its organ-specific function. In the heart, the ferritin heavy chain is the dominant subunit (~70%). The Authors found that in the infarcted area the message for the ferritin light chain is the dominant component. The change in the composition of ferritin must reflect an important change in the function of this ‘new’ ferritin.

Response:

The section in the discussion regarding the changes in ferritin light chain mRNA has been expanded to underline the importance of the results. The section now reads as follows:

“Ferritin light chain transcription has been reported to be more responsive to high iron levels than the heavy chain FTH1 (44), which was not significantly modified within infarcts. Unlike iron-storage tissues like the liver in which light chain-rich ferritin is more abundant, in the heart the ferritin heavy chain is the dominant subunit. A study in a mouse model of iron overload showed that while FTH1 mRNA levels were constitutively higher in the heart than FLC mRNA, FLC expression was more sensitive to increased cardiac iron levels (44). Increased expression of ferritin light chain in the infarct area may potentially change the composition, as well as the functional role, of cardiac ferritin following ischemia reperfusion”

6. Page 7 – “The expression of the intracellular iron sensor Iron Regulatory Protein 2 (IRP2, also known as ACO1) was significantly decreased in the infarct region (infarct, 61.4% Myo vs. remote myocardium, 102.4% Myo) (Fig. 2 C). Divalent metal transporter 1 (DMT1), which transports iron into the labile iron pool, was significantly decreased in the infarct region (infarct, 53.7% Myo vs. remote myocardium, 99.1% Myo) (Fig. 2 D). Heme oxygenase-1 (HO1) transcription is activated in response to increased intake of heme proteins, and its expression was significantly increased (infarct, 1350.3% Myo vs. remote myocardium, 100.5% Myo) (Fig. 2 E). There was increased hemoglobin staining by immunohistochemistry in reperfused and permanently occluded animals, compared to remote myocardium (Fig. 2 F,G and S7), which contributed to the increased HO1 expression.” IRP 1 binds to the stem-loop structure of the message of iron proteins; in the presence of labile iron IRP 1 changes its tertiary structure (upon

binding of ferric iron) and turns into aconitase 1. NOT IRP 2.

Response:

This mistake (IRP1/ACO1 being mislabeled as IRP2/ACO1) has been corrected throughout the text. The measured mRNA corresponds to IRP1/ACO1 (mRNA XM_003357729.3).

6. (continues) The increase of HO-1 could be associated with the increase in the level of denatured hemoglobin (metHb?), and the need to degrade the heme ‘ring’ and release its iron.

Response:

This aspect has been incorporated into the discussion text, which now reads as follows:

“Significantly increased HO1 within infarct regions suggests cellular clearance of extracellular hemoglobin and subsequent increased denatured hemoglobin requiring heme ring degradation and iron release.”

6. (continues) The numbers depicted along the entire manuscript do not represent the ‘real’ degree of significance of the measurements; i.e., 1350.3% increase. Do the Authors mean that their accuracy is valid across 5 digits?

Response:

We have corrected the numbers of expression measurements removing the decimal values.

7. Page 12 – “Elevated magnetic susceptibility shown on QSM was associated with accumulation of post-infarction tissue iron, increased labile iron ...” This association was not established.

Response

This section now read as follows:

*“Elevated magnetic susceptibility shown on QSM was associated with accumulation of post-infarction tissue iron, increased **desferrioxamine-chelatable** iron and active cellular iron homeostasis response, which were independently validated with histology and inductively coupled plasma optical emission spectrometry, electron paramagnetic resonance and gene expression analysis by real-time PCR.”*

8. Page 14 – ‘iron responsible elements’- should be iron responsive elements.

Response

This has been corrected.

9. Page 24 – “Electron Paramagnetic Resonance spectroscopy (EPR)”. The procedure described is intended to convert all the iron in the sample into ferric (3+) iron, through oxidation of all iron by desferrioxamine B. Only comparison with O-phenanthroline could allow for the quantitation of the ferric and ferrous iron.

Response:

This is correct. The section of the methods describing the EPR technique has been expanded to explain this aspect, and now reads as follows:

“Electron Paramagnetic Resonance spectroscopy (EPR)

EPR was performed with a modified method based on the study by Yegorov et al (56). Frozen samples (30-90 mg) were weighed and homogenized in 500µl of sterile saline with a TissueRuptor (Thermo-Fisher). Aliquots were saved for colorimetric iron measurements (Sigma MAK025). Decreasing concentrations of a stock solution of 10mM FeSO₄ in 0.2mM HCl (100µM-12.5µM) were diluted in saline to be used as standards. 200µl of each homogenate and FeSO₄ standards were treated for 16h with 200µl of 10mM desferrioxamine (Sigma). Desferrioxamine oxidizes Fe²⁺ to Fe³⁺, hence the desferrioxamine-bound iron detected by EPR corresponds to both original Fe³⁺ in the homogenate and Fe²⁺ converted into Fe³⁺. For EPR measurements, samples, standards and saline as blank (80 µl) were placed in 1/8" teflon tubes (Cole-Parmer) and frozen in liquid nitrogen (77 K)

Reviewer #4 (Remarks to the Author):

The authors have addressed the concerns raised by the previous reviewers in detail with additional experiments and clarifications. This is one of the few works where QSM measurements are independently validated and the authors are to be recommended for their study. I have only one minor comment.

1. (Response to Reviewer #1, Comment 13) The method used by the authors to acquire QSM in patients is indeed susceptible to misregistration between slices each acquired in a single breath-hold as the reviewer suggests. This is also the cited reason in Wen et al. (cited in the response and the manuscript) why 36% of their patients had uninterpretable QSM quality. The authors may want to add this as a potential limitation. The same group has recently presented follow-up work using a free-breathing acquisition.

The following content was added to the discussion section of the manuscript:

“Misregistration between slices was a potential source of artifacts found in the patient QSM images (Wen, Y. doi: 10.102/mrm.26808). To mitigate this issue, we minimized the breath-hold time for each slice (<20 sec) to limit cardiac and respiratory motion across the cardiac slice direction. Free-breathing cardiac-gated GRE acquisitions using respiration gating may have the potential to further improve image quality (Wen Y. doi: 10.1186/s12968-019-0579-7)”.

REVIEWERS' COMMENTS:

Reviewer #2 (Remarks to the Author):

Reviewer 2 read the revised manuscript and the detailed "Letter of Response" from the Authors. These documents are accepted as a satisfactory response to the points raised in the two previous steps of the reviewing process. This paper is now 'accepted for publication' by this reviewer.

Response to Reviewers

REVIEWERS' COMMENTS:

Reviewer #2 (Remarks to the Author):

Reviewer 2 read the revised manuscript and the detailed "Letter of Response" from the Authors. These documents are accepted as a satisfactory response to the points raised in the two previous steps of the reviewing process. This paper is now 'accepted for publication' by this reviewer.

We thank the reviewer for their comments.